# Using single-plant-omics in the field to link maize genes to functions and phenotypes

Daniel Felipe Cruz[1,2,†], Sam De Meyer[1,2,†], Joke Ampe[1,2], Heike Sprenger[1,2], Dorota Herman[1,2], Tom Van Hautegem[1,2], Jolien De Block[1,2], Dirk Inzé[1,2], Hilde Nelissen[1,2] & Steven Maere[1,2,*] iD

## Abstract

**Most of our current knowledge on plant molecular biology is based on experiments in controlled laboratory environments. However, translating this knowledge from the laboratory to the field is often not straightforward, in part because field growth conditions are very different from laboratory conditions. Here, we test a new experimental design to unravel the molecular wiring of plants and study gene–phenotype relationships directly in the field. We molecularly profiled a set of individual maize plants of the same inbred background grown in the same field and used the resulting data to predict the phenotypes of individual plants and the function of maize genes. We show that the field transcriptomes of individual plants contain as much information on maize gene function as traditional laboratory-generated transcriptomes of pooled plant samples subject to controlled perturbations. Moreover, we show that field-generated transcriptome and metabolome data can be used to quantitatively predict individual plant phenotypes. Our results show that profiling individual plants in the field is a promising experimental design that could help narrow the lab-field gap.**

**Keywords** field trial; lab-field gap; predictive modeling; single-plant -omics; *Zea mays*

**Subject Category** Plant Biology

**Mol Syst Biol. (2020) 16: e9667**

## Introduction

Efforts to develop crops with higher yield and higher tolerance to environmental stress are more important than ever in the quest for global food security and sustainable agriculture. Crop improvement increasingly relies on the identification of genes and genetic variants that impact agronomically important traits, so that beneficial variants can be engineered into the crop or incorporated in breeding programs. Mapping of quantitative trait loci (QTLs), genome-wide association studies (GWAS), and genomic prediction techniques are some of the currently preferred means of identifying the genes and variants influencing a phenotypic trait (Desta & Ortiz, 2014; Korte & Farlow, 2013). All are based on associating genetic variants, mostly single-nucleotide polymorphisms (SNPs), to observed traits in a genetically diverse population of the targeted plant species, e.g., a panel of accessions or a panel of inbred crosses between two or more parental lines (recombinant inbred lines, RILs).

Although fairly successful in some plant species, e.g., maize, these techniques also have limitations. They can only detect loci that display genetic variation in the mapping population. In addition, their resolving power is limited by linkage disequilibrium (LD), i.e., the non-random association between markers due to genetic relatedness in the population (Brachi *et al*, 2011; Huang & Han, 2014; Korte & Farlow, 2013). As a consequence, loci can often not be resolved to the individual gene level. GWA studies also have low power for rare alleles and alleles with small effect sizes, which often account for a substantial proportion of phenotypic variation, in particular for complex traits such as yield. Moreover, when mapping genotypes straight to phenotypes, the many intermediate molecular layers that articulate the phenotype from the genotype, such as the transcriptome or metabolome, are ignored. Consequently, little mechanistic insight is gained from GWAS or genomic prediction studies into how a trait is established.

As many variants uncovered in GWA studies appear to be regulating gene expression (Li *et al*, 2012; Xiao *et al*, 2017), recent efforts have sought to complement GWAS with transcriptome-wide association studies (TWAS), i.e., mapping gene expression to phenotypes in a genetically diverse population (Harper *et al*, 2012; Havlickova *et al*, 2018; Koprivova *et al*, 2014; Kremling *et al*, 2019; Pasaniuc & Price, 2017). Similarly, several recent studies have used transcriptomic or metabolomic prediction in addition to genomic prediction to associate genes to plant traits, in particular in maize (Azodi *et al*, 2020; Guo *et al*, 2016; Schrag *et al*, 2018). Azodi *et al* (2020) found that transcript levels and genetic marker data have comparable performance for predicting maize phenotypes and that performance increased when combining both data layers in a joint model. However, the use of transcriptomes and other intermediate data

---

1 Department of Plant Biotechnology and Bioinformatics, Ghent University, Ghent, Belgium
2 VIB Center for Plant Systems Biology, Ghent, Belgium
*Corresponding author. Tel: +32 93313805; E-mail: steven.maere@ugent.vib.be
†These authors contributed equally to this work as first authors

 

layers to aid genotype–phenotype mapping generally remains under-explored (Baute *et al*, 2015, 2016; Kremling *et al*, 2019).

Whereas GWAS and related methods exploit the natural genetic variation within a species to associate genes with phenotypes, systems biology studies use controlled perturbations, either genetic, environmental, or chemical, in a specific genetic background to unravel the molecular wiring of plant traits. Since the advent of high-throughput gene expression profiling platforms, massive amounts of data have been generated on the transcriptomic responses of, e.g., *Arabidopsis thaliana* Col-0 to various mutations and environmental stresses, with the purpose of unraveling the molecular processes underlying a variety of traits. However, many independent perturbations are needed to accurately reconstruct the molecular network underlying a complex trait, and no datasets exist in which any particular complex plant trait is systematically assessed molecularly and phenotypically under a large enough set of perturbations to unravel more than fragments of its molecular wiring.

The identification of a sufficient set of controlled perturbations informative of a process of interest is one of the major bottlenecks in present-day systems biology. It is often practically infeasible to identify, let alone implement, a large enough number of different controlled perturbations (mutants, stresses) relevant to a trait of interest in a single plant lineage (in contrast to GWA studies, where the genetic differences across lineages function as perturbations). Another issue is that such controlled perturbations are mostly applied in a laboratory environment, where apart from the imposed perturbation all other parameters are kept optimal and do not restrict plant growth and development. This situation does not reflect realistic field conditions, where at any given time plants are exposed to a combination of different environmental stressors with highly variable temporal and spatial patterns of occurrence (Mittler & Blumwald, 2010; Thoen *et al*, 2017). Increasing evidence is pointing toward the unique character of plant molecular responses to combinations of stresses, which often have non-additive effects on the molecular and phenotypic level (Atkinson & Urwin, 2012; Barah *et al*, 2016; Cabello *et al*, 2014; Davila Olivas *et al*, 2017; Johnson *et al*, 2014; Rasmussen *et al*, 2013; Suzuki *et al*, 2014; Thoen *et al*, 2017). As a result, perturbation studies performed under controlled laboratory conditions are often of limited predictive value for phenotypes in the field (Atkinson & Urwin, 2012; Mittler, 2006; Nelissen *et al*, 2014; Nelissen *et al*, 2019; Oh *et al*, 2009). It has been advocated that to close this lab-field gap, more -omics data and associated phenotypic data should be generated on field-grown plants (Alexandersson *et al*, 2014; Nelissen *et al*, 2019; Zaidem *et al*, 2019). Several pioneering studies have already investigated how gene expression is related to environmental stimuli in the field (Nagano *et al*, 2012; Plessis *et al*, 2015; Richards *et al*, 2012). Large-scale studies relating field-generated transcriptomes to field phenotypes are however still lacking.

Here, we propose a new strategy for studying the wiring of plant pathways and traits directly in the field, involving -omics and phenotype profiling of individual plants of the same genetic background grown in the same field. Uncontrolled variations in the micro-environment of the individual plants hereby serve as a perturbation mechanism. Our expectation is that, in addition to stochastic effects, the individual plants will be subject to subtly different sets

of environmental cues, and will in response exhibit different molecular profiles and phenotypes.

It is well known that individual plants of the same inbred line may display different phenotypes even when grown under the same macro-environmental conditions (Abley *et al*, 2016; Hall *et al*, 2007; Jimenez-Gomez *et al*, 2011; Sangster *et al*, 2008). Similar observations have been made on, e.g., inbred *Drosophila melanogaster* populations (Morgante *et al*, 2015; Whitlock & Fowler, 1999). Also on the level of gene expression, substantial variability is observed in near-isogenic populations subject to the same conditions, e.g., in plants (Cortijo *et al*, 2019; Cortijo & Locke, 2020; Jimenez-Gomez *et al*, 2011), fruitflies (Lin *et al*, 2016), and mammals (Fraser & Schadt, 2010). Both phenotypic and gene expression variability among individuals of inbred populations of higher eukaryotes have mostly been investigated from the perspective of studying the "canalization" of developmental trajectories in the face of micro-environmental variability, a concept first proposed by Waddington (1942).

mRNA and protein expression variability are also observed in clonal populations of unicellular organisms, for instance, in *Saccharomyces cerevisiae* (Ansel *et al*, 2008; Blake *et al*, 2006; Blake *et al*, 2003; Nadal-Ribelles *et al*, 2019; Raser & O'Shea, 2004) and *Escherichia coli* (Elowitz *et al*, 2002), or among single cells of, e.g., mammalia (Foreman & Wollman, 2020; Raj *et al*, 2006; Raj & van Oudenaarden, 2008; Sigal *et al*, 2006). Variability in gene expression among cells grown in the same medium is mostly attributed to "noise" caused by stochastic effects, either intrinsic (i.e., specific to the gene concerned) or extrinsic (upstream) (Cortijo & Locke, 2020; Raj & van Oudenaarden, 2008; Roeder, 2018). Studies have shown that this single-cell noise has functional consequences, both beneficial, e.g., allowing bet-hedging among cells, and detrimental (Raj & van Oudenaarden, 2008; Roeder, 2018). Furthermore, it has been shown that gene expression noise propagates through molecular networks (Pedraza & van Oudenaarden, 2005) and can be used to decipher regulatory influences (Dunlop *et al*, 2008; Munsky *et al*, 2012; Stewart-Ornstein *et al*, 2012).

Analogous to single-cell noise, gene expression differences between multicellular individuals of the same genetic background and raised in the same environment may also be useful for gene network inference. The nature of the variability between individuals may however be different (less stochastic and more micro-environmental) than between cells, as much of the single-cell stochasticity is expected to be averaged out in multicellular organisms. Earlier, we found that gene expression variations among individual *Arabidopsis thaliana* plants grown under the same stringently controlled laboratory conditions contain a lot of information on the molecular wiring of the plants, on par with traditional expression profiles of pooled plant samples subject to controlled perturbations (Bhosale *et al*, 2013). The aim of this study is to investigate to what extent we can use variability between individual field-grown plants of the same line to link genes to biological processes and field phenotypes. If even gene expression variability among laboratory-grown plants contains functionally relevant information, the molecular and phenotypic variability among field-grown plants may contain a wealth of information on processes occurring in the field.

We profiled the ear leaf transcriptome, ear leaf metabolome, and a number of phenotypes for individual field-grown maize plants of the same inbred line (*Zea mays* B104), and used the resulting data

to predict the function of genes and to quantitatively predict individual plant phenotypes. We find that our single-plant transcriptome dataset can predict the function of maize genes as efficiently as traditional laboratory-based perturbational datasets. Furthermore, we show that some quantitative phenotypes, in particular leaf-related phenotypes, can be predicted fairly well from the leaf transcriptome and metabolome data generated for the individual plants. These results open perspectives for the further use of field-generated single-plant datasets to unravel the molecular networks underlying crop phenotypes and stress responses in the field.

# Results

### Field trial design and exploratory data analysis

During the 2015 growth season, 560 maize plants of the B104 inbred line were grown in a field in Zwijnaarde, Belgium (see Materials and Methods and Fig 1A). At tasseling (VT stage), the ear leaf and the growing ear were harvested for 200 non-border plants with a primary ear at leaf 16, and plant height, the number of leaves, the length and width of the ear leaf (leaf 16) blade, husk leaf length, and ear length were measured (Dataset EV1). For 60 randomly chosen plants out of these 200, the transcriptome of mature ear leaf tissue was profiled using RNA-seq. Additionally, for 50 out of those 60 plants, metabolite profiles were generated on the same samples used for transcriptome profiling. After preprocessing and filtering (see Materials and Methods), data on the levels of 18,171 transcripts and 592 metabolites in mature ear leaf tissue were obtained for 60 and 50 plants, respectively (Dataset EV1).

As plants were harvested on two different days (because not all plants reached the VT developmental stage on the same day) and RNA-seq was performed in two batches, there may be systematic effects on some plant subgroups in the molecular and phenotypic datasets. Additionally, analysis of single-nucleotide polymorphisms (SNPs) in the RNA-seq data (see Materials and Methods) revealed that two slightly different subpopulations of plants were part of the experiment (see Appendix Fig S1). The 1,377 biallelic SNPs differentiating the two subpopulations (hypergeometric test, $q \leq 0.01$) were found to cluster mainly in regions on chromosome 1 and to a lesser extent chromosome 7 (see Appendix Fig S2). Both subpopulations were found to mostly be homozygous for one allele or the other, indicating that the mother plants of both subpopulations had different chromosome versions.

The sequencing batch, day-of-harvest (DOH), and SNP subgroup effects on transcript, metabolite, and phenotype levels were quantified jointly using linear mixed-effects (LME) models (see Materials and Methods and Dataset EV2). To avoid biases in the model *P*-values caused by spatial autocorrelations in the data (see further), these models also took into account the spatial structure of the field setup. The batch, DOH, and SNP effects explained only a minor proportion of the variance for most variables, with more than 90% of the variance allocated to the LME model residuals for 44% of transcripts, 73% of metabolites, and four out of five phenotypes (Fig EV1). However, in particular the DOH effect was found to significantly affect a sizeable proportion of the variables (Appendix Table S1), notably transcripts related to photosynthesis, transcriptional regulation, and nucleosome organization (Dataset

EV3). As we aim to leverage variability between individuals for gene function and phenotype prediction, rather than systematic variability between subgroups of plants, the batch, DOH, and SNP effects were removed from all data layers before downstream analysis, i.e., all analyses were done on the LME model residuals (Dataset EV1).

After removal of the batch, DOH, and SNP effects, no distinct sample groups are expected in our data, as no differential treatments or control measures were applied to any plant subsets. Indeed, principal component analysis (PCA) on the corrected gene expression, metabolite, and phenotype data did not reveal any clear residual group structure among the samples (Fig 1B–D). Mapping of the field layout on the PCA plots does however suggest that there is spatial structure in the data (Appendix Fig S3, see also further). Despite the fact that we kept the harvesting timeframe (10:00 am-12:00 pm) as short as possible, there may also be some time-of-harvest effect in the data. In support of this hypothesis, genes identified in Lai *et al* (2020) as having a strong diurnal rhythm ($q < $ 1e-05) have on average a higher normalized CV (see Materials and Methods) in our expression dataset than weakly rhythmic genes or non-rhythmic genes (Mann–Whitney *U*-test, $P < $ 1e-67, Appendix Fig S4). This set of strongly rhythmic genes is enriched in genes involved in photosynthesis and small-molecule metabolism (Dataset EV3). The shift in median normalized CV between strongly rhythmic genes and other genes is however small compared to the range of normalized CV values across all genes, indicating that only a minor part of the expression variance in our dataset is due to diurnal effects. Furthermore, it cannot be excluded that there are other reasons or cues than diurnal rhythmicity that may cause strongly rhythmic genes to be more variably expressed in our dataset than the average gene. As the spatial autocorrelation and time-of-harvest effects do not disturb the single-plant character of the study (in contrast to, e.g., the DOH effect), we did not attempt to remove them.

Despite the absence of designed treatments in our experimental setup, we observed substantial variability in the corrected transcriptome, metabolome, and phenotype profiles of the individual plants (Fig 2A–G). Excluding the 5% lowest-expressed transcripts, transcript levels have on average a coefficient of variation (CV) of 0.2811 across plants. Metabolite levels have a CV of 0.2726 on average, and all phenotypes have a CV $\geq 0.0523$. The gene expression variability among the field-grown maize plants, as measured by the CV, was found to be 2.49 times higher for the average gene than the expression variability among individual laboratory-grown *Arabidopsis thaliana* plants in a recent study (Cortijo *et al*, 2019) (see also Appendix Fig S5). Time point ZT06 of the *A. thaliana* dataset was taken as the reference in this comparison, as it is most comparable to the harvesting time point used for the maize dataset.

Plant-to-plant variability could either be caused by technical noise, inherent stochasticity of molecular processes within the plants, residual genetic variation in the inbred line used (even after correction for population structure) or external factors such as variability in the growth micro-environment of the individual plants. The last three processes are expected to generate biologically meaningful variation that may propagate from the molecular to the phenotypic level, or vice versa. If the variability in the data is biological in nature and propagates through the molecular networks of the plant, plants with similar gene expression profiles may be expected to also have similar metabolite and phenotype profiles. Indeed, plant-to-plant distances in transcriptome, metabolome, and

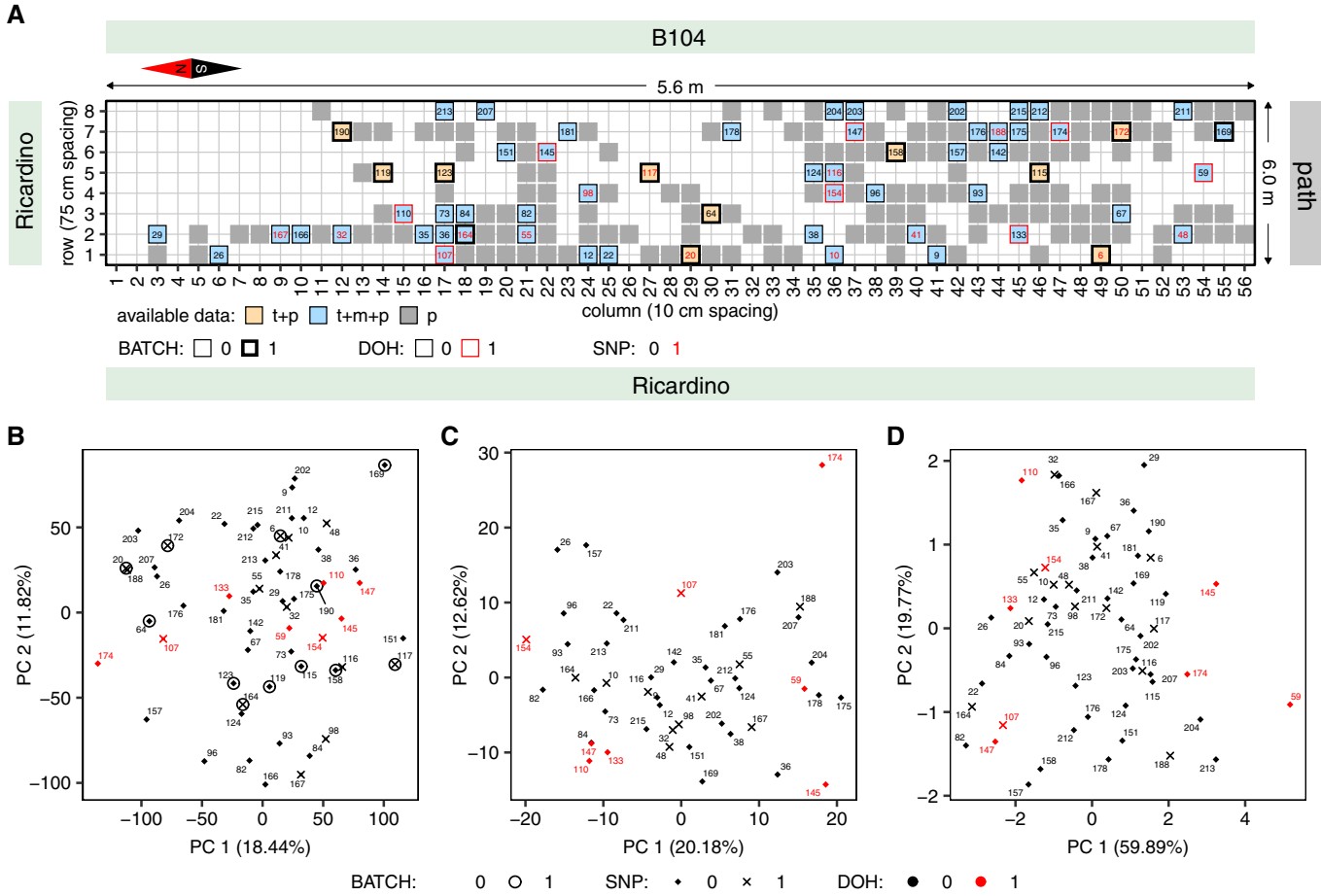

**Figure 1. Field trial design and exploratory data analysis.**

A Layout of the field trial. A total of 560 *Zea mays* B104 plants were grown in a grid of 10 rows by 56 columns. Border rows 0 and 9 are not shown on the plot, and the dimensions on the figure are not to scale. Cell colors indicate which data types are available for the plants, with gray indicating phenotype data only (p), orange transcriptome and phenotype data (t + p), and blue transcriptome, metabolome, and phenotype data (t + m + p). Harvesting dates (DOH) are indicated by the cell border color. The RNA-sequencing batch is indicated by cell border thickness. Plants belonging to different subgroups based on SNP analysis are indicated by the coloring of the plant ID numbers inside the cells. The designations 0 and 1 for the DOH, BATCH, and SNP effect groups are used for the largest and smallest group, respectively.

B Plot showing the first two principal components (PCs) in a PCA of the 60 single-plant transcriptomes.

C Plot showing the first two PCs in a PCA of the 50 single-plant metabolomes.

D Plot showing the first two PCs in a PCA of the plant phenomes for the 60 plants that were RNA-sequenced.

Data information: The plants in panels (B–D) are numbered according to the numbering in panel (A). Plants belonging to different SNP and DOH subgroups are indicated by different markers and marker colors, respectively, and plants sequenced in the second, smallest batch are circled in panel (B).

phenotype space were found to be significantly positively correlated (Fig EV2). Interestingly, the phenotype distance between plants was also significantly positively correlated with the physical distance between plants in the field. All phenotypes except ear length were found to be spatially autocorrelated at $q \leq 0.05$ (see Materials and Methods, Fig EV3 and Dataset EV2). A weak but significant positive correlation was also found between the metabolome distance and physical distance between plants, and 48 out of 592 metabolites exhibit spatial patterning at $q \leq 0.01$ (Moran's I, Dataset EV2). A borderline significant correlation was found between the physical distance of plants and their overall distance in transcriptome space (Fig EV2), indicating that most genes do not exhibit spatially patterned gene expression. However, spatial autocorrelation

analysis of the transcriptome data revealed that 2,574 out of 18,171 transcripts do exhibit spatial patterning at $q \leq 0.01$ (Moran's I, Dataset EV2). Among the transcripts and metabolites with significant spatial patterning, spatial covariance was found to make up around 60% of the residual variance in the LME models on average (see Appendix Fig S6).

The spatially autocorrelated transcripts and metabolites were grouped in 16 and 2 co-expression clusters, respectively (see Materials and Methods, Dataset EV4 and Appendix Figs S7 and S8). Significant GO enrichments were found in 9 of the autocorrelated transcript clusters, e.g., clusters 5 and 6 were found enriched in genes involved in the response to chitin, and clusters 14 to 16 in chloroplast-associated genes (Dataset EV4). This indicates that the

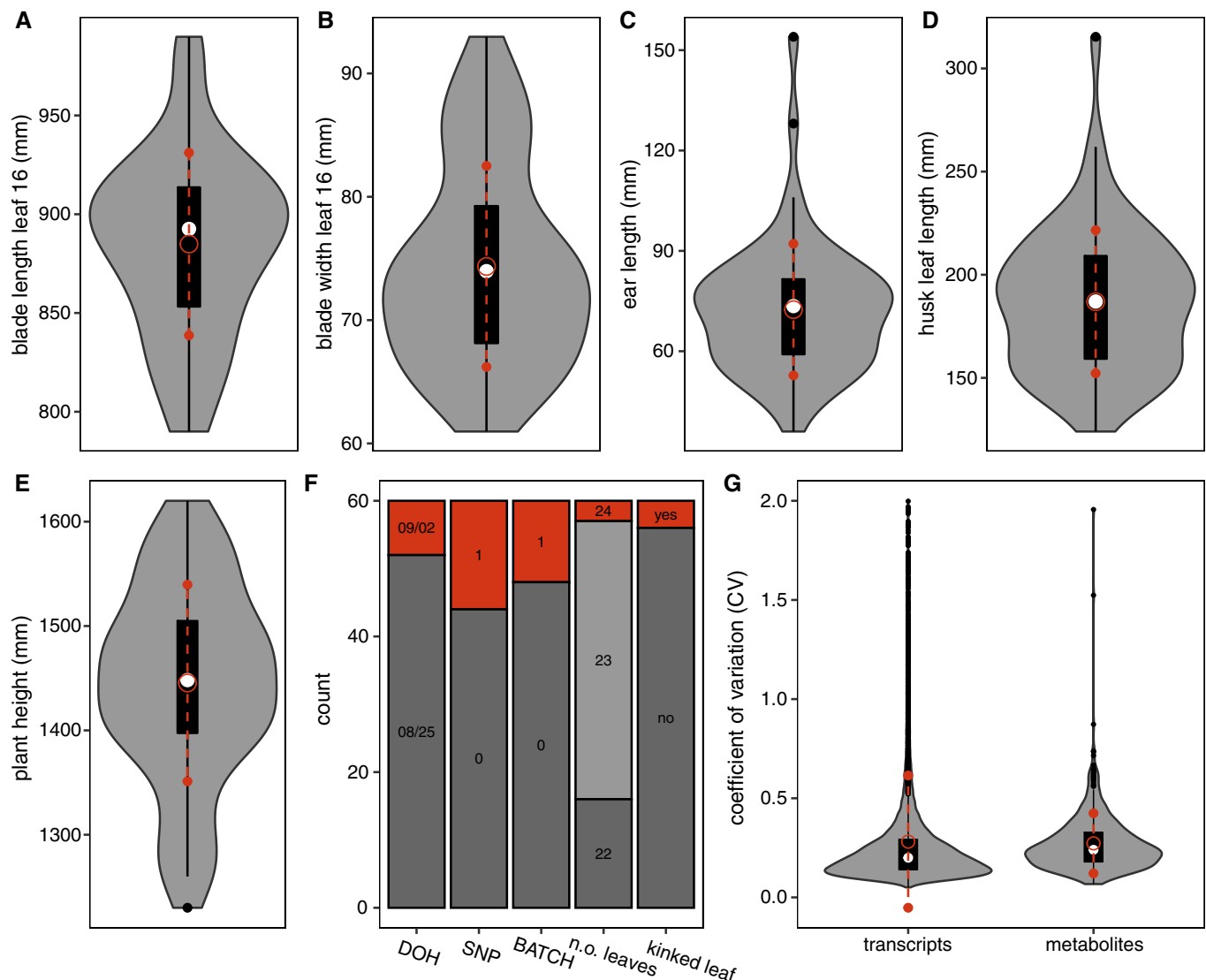

**Figure 2. Transcriptomic, metabolic, and phenotypic variability among individual field-grown maize plants.**

A–G  In panels (A) to (E), violin plots show the variability in continuous leaf 16, ear, and plant height phenotypes among the 60 individual plants used in downstream analyses. Panel (F) depicts how many of the plants were harvested on different days (DOH), belong to different SNP subgroups, or were RNA-sequenced in different batches. This panel also displays the variability in two discrete phenotypes, namely the number of leaves at harvest and whether or not leaf 16 was kinked. Panel (G) shows violin plots for the distribution of the coefficient of variance (CV) across the sampled plants for the levels of individual transcripts and metabolites. For visualization purposes, the transcript CV was capped at 2.0.

Data information: In all violin plots, the median is indicated by the white circle. The black box extends from the 25th to the 75th percentile, and black whiskers extend from each end of the box to the most extreme values within 1.5 times the interquartile range from the respective end. Data points beyond this range are shown as black dots. The red open circle indicates the mean of the distribution, with red whiskers extending to 1 standard deviation above and below the mean.

activity of several biological processes varied across the field in a spatially patterned way. Three of the 16 autocorrelated transcript clusters and both of the autocorrelated metabolite clusters correlated with at least one measured phenotype at $q \leq 0.05$ (Appendix Figs S9 and S10). The average gene expression profile of transcript cluster 2, for instance, correlates significantly with ear length (Fig 3). Interestingly, one of the genes in cluster 2 is GRMZM2G171365 (*SUPPRESSOR OF OVEREXPRESSION OF CONSTANS 1*, *ZmSOC1*, *ZmMADS1*), a MADS-box transcription factor known to promote flowering (Alter *et al*, 2016; Zhao *et al*, 2014) and also known to be upregulated in leaves during the floral transition (Alter *et al*, 2016). Overall, the presence of spatially autocorrelated patterns in the transcriptome, metabolome, and phenotype data indicates that at least part of the variability observed among the individual plants is due to micro-environmental factors that have a spatial structure. Correlations between the molecular and phenotypic data layers indicate that this variability propagates from one layer to another.

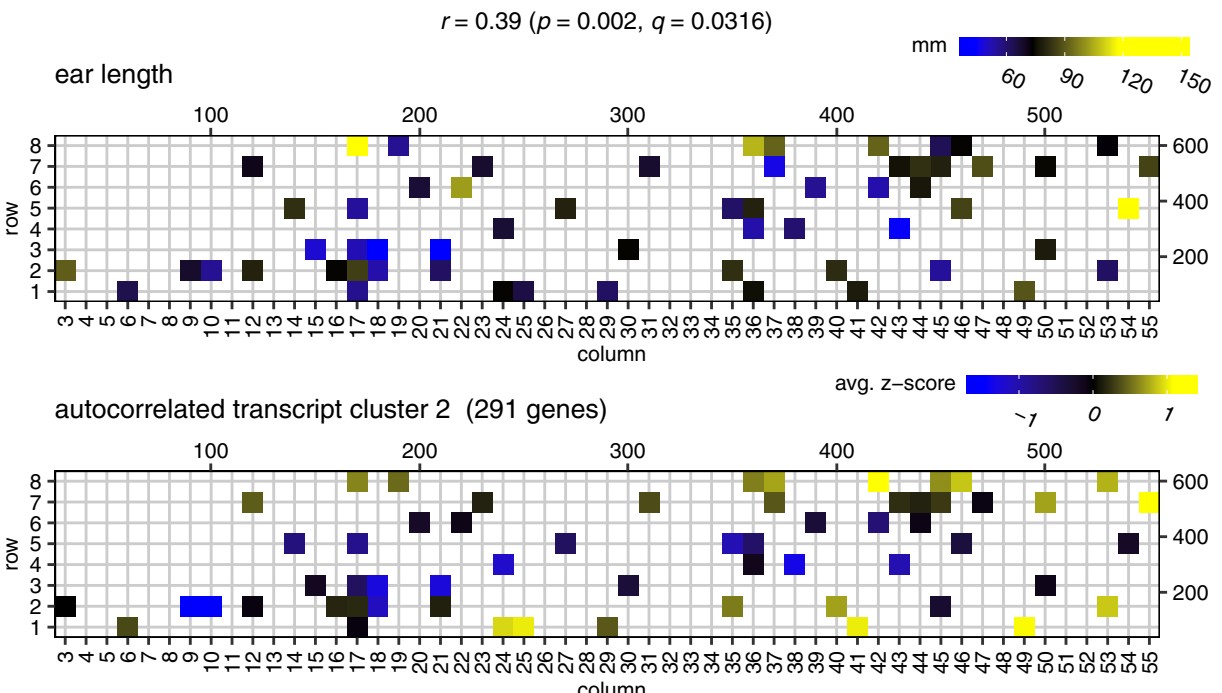

**Figure 3. Gene expression patterns in cluster 2 correlate with ear length.**

The top panel displays the ear length phenotype on the field (only for plants that were transcriptome profiled). The bottom panel displays the average z-scored gene expression profile of spatially autocorrelated gene cluster 2 (291 genes), mapped to the field. Shown on top are the Pearson's correlation (r) between the cluster 2 expression profile and ear length, the corresponding P-value (computed using cor.test in R) and the corresponding q-value (computed using the Benjamini–Hochberg method on all comparisons of cluster gene expression profiles with the ear length profile). The scales on the top and to the right of the field maps give field plot dimensions in cm.

## Variability of gene and metabolite expression across plants gives insight into biological processes active in the field

We investigated which genes have highly variable expression levels in the field setting used, and which ones are stably expressed across the field. We ranked genes based on a normalized coefficient of variation (*normCV*) of their gene expression profile across the field (see Materials and Methods and Dataset EV5), excluding the 5% lowest-expressed genes. We found that stably expressed genes have on average longer coding sequences than variably expressed genes and have on average more introns and exons (Appendix Table S2). Similar results were previously obtained in the study of Cortijo *et al* (2019) on individual *A. thaliana* plants, and the authors showed that their observations could not be accounted for by technical artefacts related to differences in the average RNA-seq coverage of longer versus shorter genes.

Mann–Whitney *U*-tests (Mann and Whitney, 1947) were performed to determine which Gene Ontology (GO) biological processes are represented more at the top or bottom of the CV-ranked gene list than expected by chance (Dataset EV6). Genes related to photosynthesis, response to biotic and abiotic stresses, cell wall organization, secondary metabolism, brassinosteroid metabolism, and response to hormones such as cytokinin, abscisic acid, jasmonic acid, and gibberellin were found to be among the more variably expressed genes across the field, suggesting that the

harvested leaves were differentially impacted by various stress factors. The processes that are most stably expressed across the field are mainly housekeeping processes related to, e.g., the metabolism and transport of proteins and mRNAs, and chromatin organization (Dataset EV6). Interestingly, the GO enrichments obtained for variably and stably expressed genes in the field-grown maize plant dataset are largely in line with the results reported by Cortijo *et al* (2019) on laboratory-grown *A. thaliana* plants. Photosynthesis, secondary metabolism, cell wall organization, abiotic stress, and defense response genes, for instance, were also found enriched by Cortijo *et al* (2019) in several of the highly variable gene sets they compiled for different sampling time points in a 24 h time span, while RNA and protein metabolism genes feature prominently in some of their lowly variable gene lists.

The metabolites in our dataset were also ranked based on their variability in abundance across the field, again based on a normalized coefficient of variation (see Materials and Methods, Appendix Fig S11, and Dataset EV5). The list of the 50 most variable metabolites mainly includes primary metabolites, in particular compounds involved in amino acid and sugar metabolism, but also secondary metabolites such as naringenin, chrysoeriol, beta-carotene, and benzoate. Among the 50 least variable metabolites, there are five dipeptides and four compounds involved in vitamin metabolism. Given the fairly limited number of identified metabolites in our dataset, distinguishing clear trends is however harder than for genes.

Hierarchical clustering of the transcriptome and metabolome data offers an overall view of the molecular variability across the plants profiled (Fig EV4). Several clusters were found to be significantly enriched in genes involved in particular biological processes, further confirming that the single-plant dataset contains biologically meaningful information (Dataset EV7). Also the biclustering approach ENIGMA (Maere *et al*, 2008) yielded a variety of modules enriched in genes involved in processes such as photosynthesis, cell wall organization, response to chitin, and others (Dataset EV7). An example ENIGMA module enriched for photosynthesis and response to light stimulus genes is shown in Fig 4. In this module and many others (see, e.g., Fig EV4), different subgroups of plants show clearly different expression profiles, highlighting that many processes are not homogeneously active across the field.

### Gene function prediction from single-plant transcriptome data

In previous work, we showed that expression variations among individual *Arabidopsis thaliana* plants, all grown under the same stringently controlled conditions, can efficiently predict gene functions (Bhosale *et al*, 2013). A complicating factor in this study however was that the individual plants profiled were of different genetic backgrounds and were grown in different laboratories (Massonnet *et al*, 2010). Although laboratory and genotype effects and

their interaction were removed from the data and the results pointed to micro-environmental or stochastic differences between plants as the main cause of the residual gene expression variability, it cannot be excluded that residual non-linear laboratory or genotype effects may have influenced the results. In this respect, the current dataset on individual maize plants of the same line grown in the same field is likely better suited to assess whether expression variations between individuals grown under the same conditions can be used to predict gene functions, despite the potential presence of remnants of other systematic effects in our data (day of harvest, sequencing batch, population substructure). The phrasing "same conditions" is to be understood here in the sense that there are no deliberate treatment differences between plants, only uncontrolled micro-environmental differences. These are likely larger in the current field setup than in the controlled laboratory setup on which the Bhosale *et al* (2013) study was based.

We constructed a network of significantly coexpressed genes from the transcriptome data, using spatially adjusted Pearson correlation coefficients between the $log_2$-transformed gene expression profiles (see Materials and Methods). Accounting for the spatial autocorrelation structure of our field-generated data is necessary to avoid inflation of the false-positive rate (Lennon, 2000). The function of any given gene in this co-expression network was predicted based on the annotated functions of the gene's network neighbors

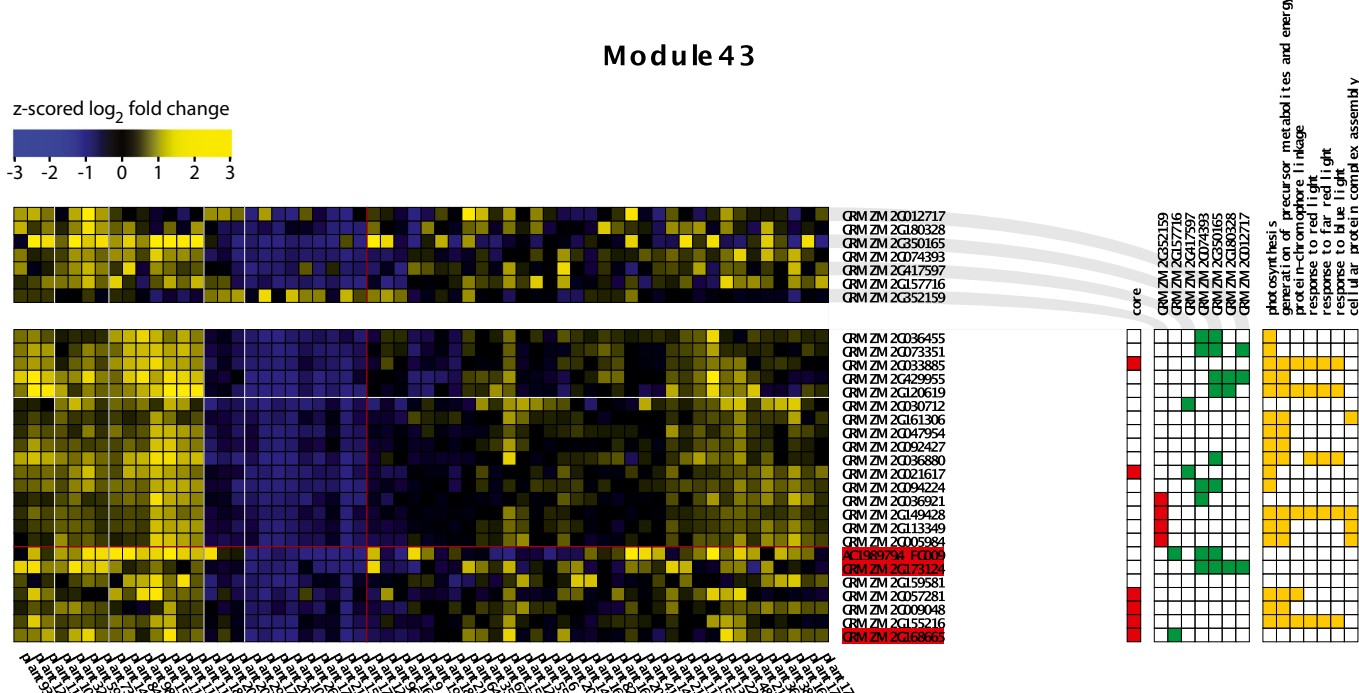

**Figure 4.  Example ENIGMA module learned from the single-plant transcriptome dataset.**

The bottom yellow/blue grid shows the expression profiles of the module genes, while the top grid contains the expression profiles of predicted regulators of the module. Yellow/blue squares indicate higher/lower gene expression with respect to the average expression of a gene across plants (black). Color hues are based on z-scoring the $log_2$ expression fold changes of genes (with respect to their average expression) across the entire dataset. Significant co-differential expression links between the regulators and the module genes are indicated in the red/green matrix to the right (green = positively correlated, red = negatively correlated). Gene names highlighted in red indicate regulators that are part of the module. Genes indicated as core genes belong to the original module seed, and other genes were accreted by the seed in the course of module formation (Maere *et al*, 2008). Enriched GO categories in the module gene set are displayed on the right, with orange squares depicting which module genes are annotated to these GO categories. This particular module is significantly enriched ($q \leq 0.01$) in known photosynthesis genes.

**Table 1.  Topological parameters for the single-plant network and the networks sampled from the SRA and diversity panel datasets. The "predicted positives" column indicates the amount of true-positive plus false-positive predictions made by each type of network at $q \leq 0.01$**

| | # Nodes | # Edges | Network density | Average clustering coefficient | Unannotated gene fraction | Predicted positives |
|---|---|---|---|---|---|---|
| Single-plant network | 10,501 | 878,079 | 0.015927 | 0.481209 | 0.085135 | 291,237 |
| SRA networks mean | 10,256 | 878,079 | 0.017302 | 0.478519 | 0.091104 | 169,571 |
| SRA networks SD | 1,158 | 0 | 0.003674 | 0.022180 | 0.003826 | 25,649 |
| *P*-value single-plant versus SRA networks | 0.377 | – | 0.377 | 0.465 | 0.052 | 0.004 |
| Diversity networks mean | 14,193 | 878,079 | 0.013921 | 0.323074 | 0.102636 | 111,555 |
| Diversity networks SD | 5,177 | 0 | 0.010709 | 0.125699 | 0.008788 | 46,866 |
| *P*-value single-plant versus diversity networks | 0.436 | – | 0.574 | 0.039 | 0.001 | 0.001 |

(see Materials and Methods). To compare the function prediction performance of our single-plant dataset with that of traditional gene expression datasets on pooled samples of plants grown under controlled conditions, we ran the same function prediction pipeline on 500 networks constructed from gene expression datasets on maize leaves available from the Sequence Read Archive (SRA) transcriptome database (see Materials and Methods and Dataset EV8). Each of these 500 networks was inferred from a dataset of the same size as the single-plant dataset, containing 60 transcriptome profiles sampled from the SRA. The number of significant edges (Bonferroni-corrected $P \leq 0.01$) inferred from these sampled datasets was systematically higher than the number of edges inferred from the single-plant dataset. One factor causing this is that the SRA transcriptome data exhibit clear groups of experimental conditions for which expression profiles are more similar within groups than between groups (Appendix Fig S12), more so than the single-plant data. This group structure causes inflated correlation $P$-values in the sampled networks. Since the function prediction performance of correlation networks is dependent on the amount of edges included (Appendix Fig S13), the number of edges in each sampled network was fixed to the number of significant edges observed in the single-plant network (878,079 edges). Other network properties such as the number of nodes, network density, and average clustering coefficient are not significantly different between the resulting sampled networks and the single-plant network, but the single-plant network does contain slightly less genes of unknown function than the average sampled network (Table 1).

The overall gene function prediction performance of all networks was scored using known GO annotations for maize as the gold standard (see Materials and Methods). For each network, we calculated the fraction of known gene function annotations recovered by the predictions (recall), the fraction of gene function predictions supported by the gold standard (precision), and the F-measure (harmonic mean of precision and recall) at different false discovery rate (FDR) levels, ranging from $q = 0.01$ to $10^{-11}$ (Fig 5A–D). Except at the least stringent prediction threshold ($q = 10^{-2}$), the recall of the single-plant network was higher than the 75th percentile of the recall values for the SRA sampled networks, indicating that the single-plant network predictions generally recover more known gene functions than the sampled network predictions. On the other hand, the predictions of the single-plant network are generally less precise than

those of the average sampled network, except at lower-confidence prediction thresholds ($q \geq 10^{-4}$). The overall function prediction performance of the single-plant network (as measured by the F-measure) is higher than the 75th percentile of the SRA networks for most of the $q$-value range, except for $q \leq 10^{-10}$. The comparatively lower F-measures for $q \leq 10^{-10}$ are mostly due to the lower precision of the single-plant network predictions at higher confidence levels compared to the sampled networks, indicating that a bigger proportion of the high-confidence function predictions made by the single-plant network is not supported by the gold standard.

There are reasons to believe that not all of these excess false-positive predictions made by the single-plant network at high confidence levels are truly wrong. First, the GO annotation for maize, used here as the gold standard, is incomplete. Of the 39,479 genes in the maize B73 reference genome annotation (AGPv3.31), 9,884 have no biological process assignments in the GO annotation file we compiled (see Materials and Methods), and many others likely have incomplete or faulty annotations (Rhee & Mutwil, 2014; Wimalanathan *et al*, 2018). High-confidence gene function predictions labeled as false positives may therefore be regarded rather as new gene function predictions to be tested. By itself however, the incompleteness of the gold standard should not lead to a specific disadvantage for the single-plant network, as all networks are compared on the same footing. More importantly, the current annotations in GO are mostly derived from traditional laboratory-based perturbation experiments on pooled plant samples, akin to the ones used to construct the sampled networks. This may create a bias in favor of the sampled networks, in particular for the precision measurements (see also Discussion). The recall measure should therefore probably get a higher weight when comparing the gene function prediction performance of the single-plant and sampled networks.

The analysis outlined above compares the gene function information content of expression data generated on individual field-grown plants versus data generated on pooled plant samples subject to controlled treatments. In both cases, the plants profiled come from a single inbred line. To assess how the information content of expression data on individuals of a single line compares to that of expression data on a diversity panel as used for GWAS and TWAS, we performed the same analysis on 100 mature leaf gene expression compendia sampled from a recent diversity panel dataset (Kremling *et al*, 2018) (see Materials and Methods, Table 1 and Fig EV5).

Some of the patterns observed are similar to those observed in the comparison with SRA datasets, namely that the single-plant dataset generates more function predictions than the diversity datasets, but with lower precision over most of the prediction $q$-value range, in particular for higher confidence levels. The recall values for the single-plant dataset on the other hand are systematically higher than for the sampled diversity datasets. As a result, the gene function prediction performance ($F$-measure) of the single-plant dataset is higher than that of all sampled diversity datasets.

## Single-plant dataset contains information on biological processes that are active and varying between plants in the field context

To assess whether the single-plant dataset contains more information on some biological processes than on others, we investigated how well the gene function predictions on the single-plant network and sampled SRA networks could recover the genes involved in specific biological processes (see Materials and Methods). The function prediction performance of all networks was scored for 207 different GO categories, including the categories investigated in Bhosale *et al* (2013) (Dataset EV9). Figure 6A–D shows the relative performance of the single-plant network for a selection of GO categories related to abiotic and biotic stress responses, hormonal responses, and development (see Dataset EV9 and Appendix Fig S14 for results on other GO categories).

For abiotic stresses, the single-plant network scores very well compared to the sampled SRA networks for responses to cold and heat, salt stress, and drought (water deprivation), all of which are relevant from a field perspective (Fig 6A). For light responses, the picture is more nuanced, with very good performance for response to UV light, average performance for response to blue light, ambiguous performance for categories related to "*response to red- and far-red light*" and very poor performance for "*response to light intensity*" and "*photoperiodism*". The overall very good function prediction performance for "*response to abiotic stimulus*" indicates that there is considerable variation across the field in the transcriptional activity of the genes concerned, which suggests that the individual plants were subject to multiple abiotic environmental cues that varied in intensity across the field.

Concerning responses to biotic stimuli, the single-plant predictions score very well for the "*response to herbivore*" and "*response to bacterium*" categories, but poor for responses to fungi, nematodes, viruses, and symbionts (Fig 6C, Dataset EV9 and Appendix Fig S14). This indicates that the individual plants may have been variably exposed to bacteria and herbivores in particular. The single-plant network also scored very well for some GO categories related to biotic stimulus responses that are not shown in Fig 6C, such as "*defense response*" and "*response to chitin*" (Dataset EV9 and Appendix Fig S14). The function prediction performance for other biotic stress categories such as "*response to insect*" or "*response to oomycetes*" could not be assessed because both the sampled and single-plant datasets did not yield enough predictions (see Materials and Methods).

Similarly, both the sampled and single-plant datasets failed to deliver sufficient predictions to score the function prediction performance for responses to ethylene, gibberellins, salicylic acid, and strigolactones (Fig 6D). Among the hormone responses for which the gene function prediction performance of the single-plant dataset

could be scored, the responses to abscisic acid (ABA), cytokinin, and jasmonic acid score very well, "*response to brassinosteroids*" scores average and "*response to auxin*" scores very poorly. The very poor function prediction performance for auxin response genes is consistent with the fact that only mature leaf tissue was profiled in the single-plant experiment, where auxin signaling is less active (Brumos *et al*, 2018). In contrast, the sampled datasets also contain experiments on entire leaves, leaf primordia, and leaf zones such as the division and elongation zone where auxin signaling is more active (Dataset EV8).

Regarding developmental processes, the single-plant dataset scores very well for predicting genes involved in leaf development and embryo development, and very poor for flower, fruit, and seed development (Fig 6B). The very good prediction performance for embryo development may come as a surprise given that only leaf material was profiled, but one needs to keep in mind that all performances are scored relative to the performance of the sampled SRA datasets, which also exclusively profiled leaves. Even then, it may be considered surprising that leaf expression profiles contain any information at all on developmental processes occurring in other tissues. However, aspects of development may be shared across tissues. Several root development genes, for instance, were found to also function in some capacity in leaves (Taniguchi *et al*, 2017; Yang *et al*, 2019). The developmental program of leaves may overlap with that of embryos in particular as the latter also contain embryonic leaves. More genuinely surprising is that the single-plant dataset outperforms more than 75% of the sampled SRA datasets for predicting genes involved in leaf development, both in terms of precision and recall, despite only profiling mature leaf tissue of ear leaves.

## Exploration of new maize genes predicted to be involved in biotic and abiotic stress responses

In total, 1,334,456 novel gene function predictions (i.e., predictions not matching GO annotations) were obtained from the single-plant dataset at $q \leq 0.01$ (Dataset EV10). To assess the quality of these predictions, we performed a literature screen to search for evidence supporting the top-10 regulator predictions for the GO categories "*response to chitin*", "*response to water deprivation*", and "*C₄ photosynthesis*". The first two are categories for which the single-plant dataset exhibited very good gene function prediction performance compared to the sampled SRA datasets. "*C₄ photosynthesis*" on the other hand scored very poorly in the single-plant dataset (Dataset EV9 and Appendix Fig S14). We included this category in the literature validation effort to assess whether poor gene function prediction performance for a biological process, as scored based on which genes are already annotated to the process in GO, also entails that newly predicted links between genes and the process under study are of poor quality.

"*Response to chitin*" was among the best-scoring GO categories in our assessment of the gene function prediction performance of the single-plant dataset. Chitin is a main component of fungal cell walls and insect exoskeletons (Fleet, 1991; Latgé, 2007), and the response to chitin is therefore closely related to the responses to fungi and insects. For three out of the top-10 novel transcriptional regulators predicted to be involved in the response to chitin (Appendix Table S3), we found indirect evidence in literature in

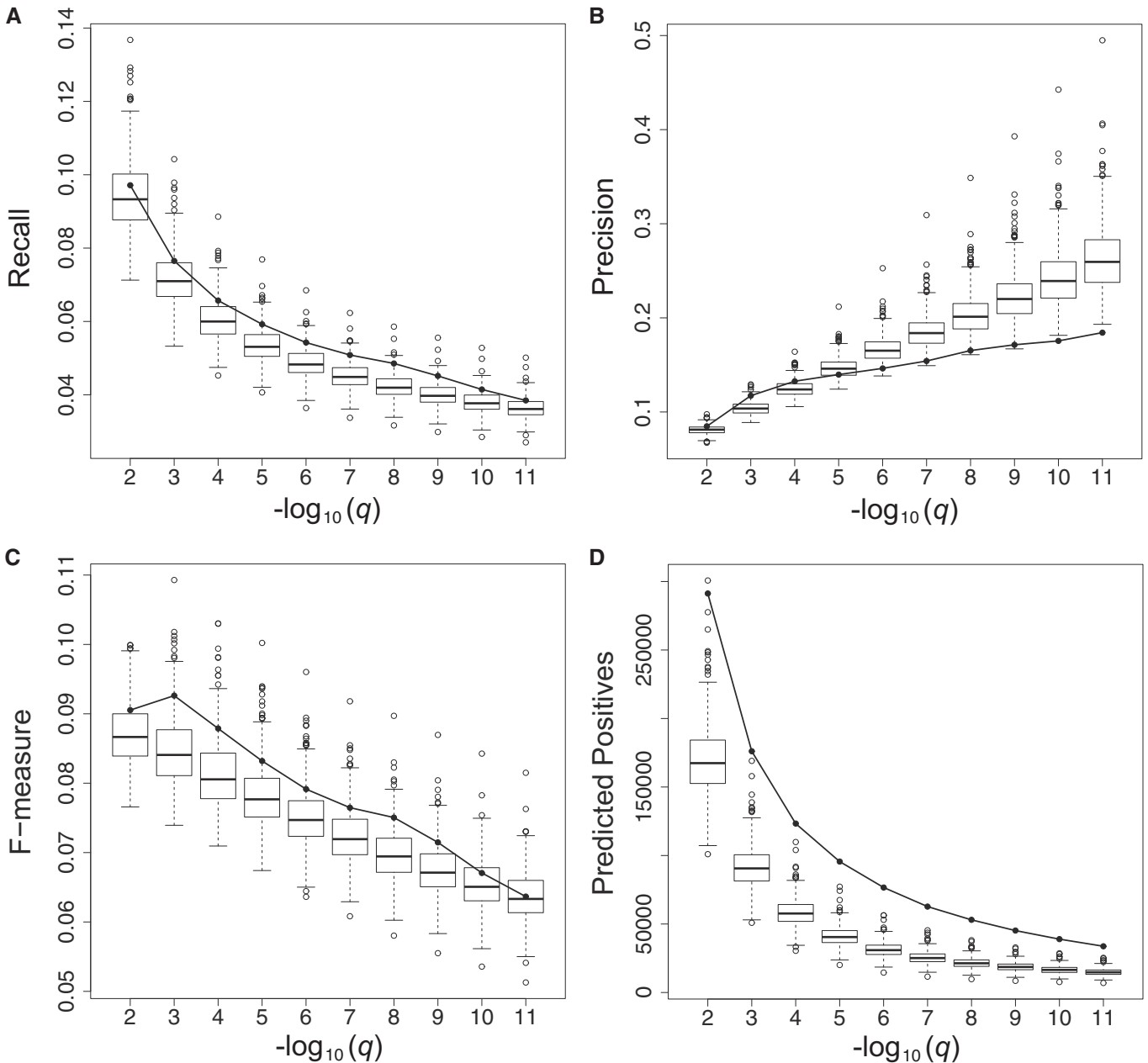

**Figure 5. Global gene function prediction performance.**

A–D   Panels (A) to (D) depict the gene function prediction performance of the single-plant network (solid line) and 500 sampled SRA networks (box-and-whisker plots) averaged across all genes in a given network. Boxes extend from the 25th to the 75th percentile of the sampled networks, with the median indicated by the central black line. Whiskers extend from each end of the box to the most extreme values within 1.5 times the interquartile range from the respective end. Data points beyond this range are displayed as open black circles. Panels (A), (B), and (C), respectively, represent the recall, precision, and F-measure of the network-based gene function predictions as a function of the prediction FDR threshold ($q$). Panel (D) depicts the number of gene functions predicted from each network (predicted positives = true positives + false positives) as a function of the prediction FDR threshold. As multiple gene functions can be predicted per gene, the number of predicted positives is generally higher than the number of genes.

support of the predictions. *ZmWRKY53* (GRMZM2G012724), on the 1[st] position in the ranking, was previously found to be involved in the response of maize to *Aspergillus flavus,* a fungal pathogen that affects maize kernel tissues and produces mycotoxins that are harmful for humans and animals (Fountain *et al*, 2015). *ZmWRKY53* was found to be strongly upregulated in both a susceptible and a

resistant maize line upon inoculation of kernels with *Aspergillus flavus* (Fountain *et al*, 2015). Its putative functional ortholog in *Arabidopsis thaliana*, *AtWRKY33*, is known to regulate defense response genes (Birkenbihl *et al*, 2012; Zheng *et al*, 2006), and its putative functional orthologs in *Triticum aestivum* (*TaWRKY53*) and *Oryza sativa* (*OsWRKY53*) have previously been suggested to

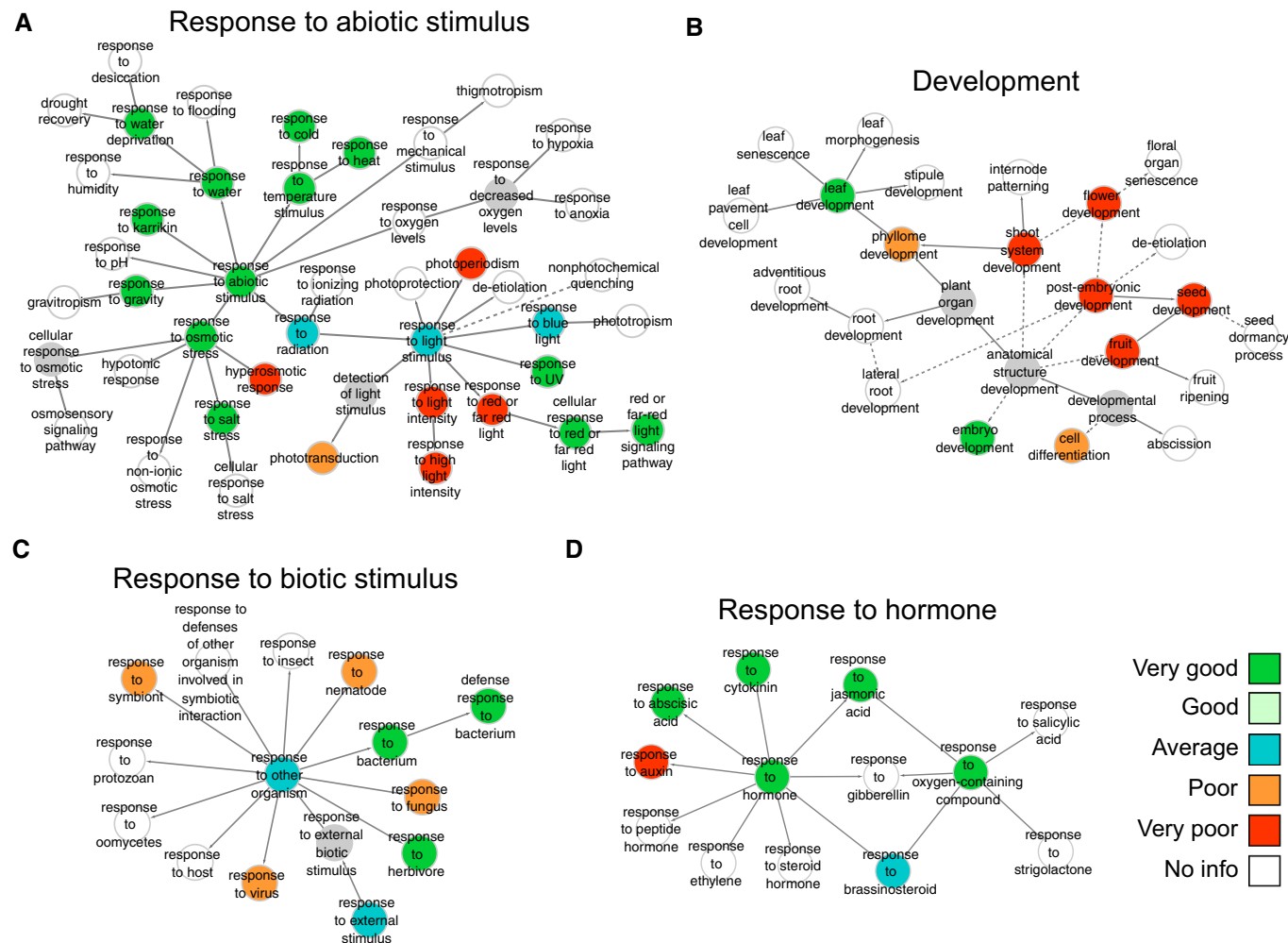

**Figure 6. Gene function prediction performance for specific GO categories.**

A–D   Panels (A) to (D) show the gene function prediction performance of the single-plant network versus sampled SRA networks for GO categories related to abiotic stimulus responses, development, biotic stimulus responses, and hormone responses, respectively. Categories are shown in the context of the GO hierarchy and colored according to how well the single-plant network performs in comparison with 500 sampled SRA networks (see Materials and Methods). Solid arrows represent direct parent–child relationships in GO, and dashed arrows represent indirect relationships. Gray nodes depict untested GO categories. White nodes depict GO categories for which there was insufficient information to score the performance of the single-plant network versus the sampled networks, i.e., categories for which the single-plant network and more than half of the sampled networks did not give rise to any predictions at $q \leq$ 1e-2.

regulate several biotic and abiotic stress response genes, including chitinases (Van Eck *et al*, 2014). Overexpression of *OsWRKY53* was also shown to increase the resistance of *O. sativa* to herbivory by the brown planthopper *Nilaparvata lugens* (Hu *et al*, 2016). Another WRKY TF in the top-10 list, *ZmWRKY92* (GRMZM2G449681, rank 7), was previously found to be induced upon *Fusarium verticillioides* inoculation of kernels in the ear rot-resistant maize inbred line BT-1 (Wang *et al*, 2016). Yet another WRKY TF, *ZmWRKY14* (GRMZM2G091331, rank 8), is orthologous to *AtWRKY15* (*AT2G23320*), a known chitin-responsive TF in *A. thaliana* (Libault *et al*, 2007). Two other genes in the top-10 list are linked to defense responses, but have not been linked specifically to the response to chitin: GRMZM2G027958 (rank 6), a putative *BRASSINOSTEROID INSENSITIVE 1-associated* receptor kinase whose *A. thaliana* ortholog AT2G31880 (*EVERSHED, EVR, SOBIR1, SUPPRESSOR OF*

*BIR1 1*) regulates cell death and defense responses (Albert *et al*, 2015; Gao *et al*, 2009), and GRMZM2G106792 (rank 9), a gene homologous to *NDR1/HIN1-like* genes in *A. thaliana*, most of which are induced upon particular viral (Zheng *et al*, 2004) or bacterial (Varet *et al*, 2002) infections.

The second GO category for which we screened literature is "*response to water deprivation*". Seven of the top-10 transcriptional regulators predicted to be involved in drought stress responses, but not annotated as such in GO, have previously been linked to drought stress in other studies (Appendix Table S4). *ZmXLG3b* (GRMZM2G429113, rank 1), encoding a guanine nucleotide-binding protein predicted to be involved in the response to desiccation, was previously found to be downregulated in the drought-tolerant H082183 line but upregulated in the drought-susceptible maize line Lv28 under severe drought stress versus control conditions (Zhang

et al, 2017). Moreover, *ZmXLG3b* was identified as a candidate drought stress response gene in a GWAS study on 300 inbred maize lines, and its expression level was found to anticorrelate with drought stress tolerance levels in four tested maize lines (Yuan et al, 2019). *ZmMPK3-1* (GRMZM2G053987, rank 2), a mitogen-activated protein kinase (MAPK), was previously found to be upregulated in leaf and stem tissue upon drought stress in maize (Liu et al, 2015b). Furthermore, the top-10 contains 3 bZIP and 2 NAC transcription factors with drought stress-responsive expression profiles. In a recent study (Cao et al, 2019), *ZmbZIP111* (GRMZM2G073427, rank 3) was found to show decreased expression under polyethylene glycol (PEG)-induced drought stress and a sharp increase in expression upon rewatering, and *ZmbZIP9* (GRMZM2G092137, rank 5) was found to exhibit the opposite behavior. Similarly, *ZmNACTF77* (AC196475.3_FG005, rank 9) was found to show increased expression under PEG-induced drought stress and a sharp decrease in expression upon rewatering, while *ZmNACTF53* (GRMZM2G059428, rank 4) was found to show a temporary sharp decrease in expression under PEG-induced drought stress (Wang et al, 2020). Expression of *ZmbZIP60* (GRMZM2G025812, rank 8) was also found to be rapidly and strongly induced by dehydration (Wang et al, 2012).

Finally, we screened literature for the top-10 regulators predicted to be involved in $C_4$ photosynthesis (Appendix Table S5). Surprisingly, the single-plant dataset performed very poorly for the light-associated GO categories "photosynthesis" and "$C_4$ photosynthesis" (Dataset EV9 and Appendix Fig S14), even though several "response to light stimulus" subcategories scored very well (Fig 6A) and though our clustering analyses revealed several (bi)clusters heavily enriched in photosynthesis genes (see Dataset EV7). The performance plots show that the very poor function prediction performance for photosynthesis categories is due to the single-plant predictions having a very low precision compared to the predictions from the sampled SRA datasets, while the number of predictions made by the single-plant data and their recall are comparatively very high (Appendix Fig S14). As argued above, recall values may be more indicative for the quality of gene function predictions than precision values, given the incompleteness of the maize GO annotation we use as a reference. If this is the case, genes that are predicted with high confidence to be involved in $C_4$ photosynthesis but were scored as false positives by GO may still offer valuable leads. Indeed, we found evidence in literature linking four of the top-10 predicted regulators to $C_4$ photosynthesis. *ZmCSP41A* (GRMZM2G111216, rank 1), a highly conserved sequence-specific chloroplast mRNA binding protein and unspecific endoribonuclease, was previously found to be more highly expressed in bundle sheet chloroplasts than in mesophyll chloroplasts (Friso et al, 2010). In the genus *Flaveria*, which contains $C_3$ and $C_4$ species as well as intermediates, a homolog of *ZmCSP41A* was found to be downregulated in leaves of $C_4$ species compared to $C_3$ species (Gowik et al, 2011). Transcripts of *ZmCRB* (GRMZM2G165655, rank 2) also accumulate preferentially in bundle sheet cells and are known to stabilize several chloroplast transcripts, e.g., for photosystem I and II components (John et al, 2014). *ZmbHLH32* (GRMZM2G180406, rank 7) is orthologous to *A. thaliana* CRYPTOCHROME INTERACTING BASIC-HELIX-LOOP-HELIX (CIB) genes, known to regulate photosynthesis, and *ZmbHLH32* transcripts have been shown to preferentially accumulate in bundle sheath cells, while transcripts of other maize CIB orthologs preferentially accumulate in mesophyll

cells (Hendron & Kelly, 2020). *ZmSIG5* (GRMZM2G543629, rank 8) encodes a plastid sigma factor. Several homologous sigma factors in the *Flaveria* and *Cleome* genera were found to be upregulated in leaves of $C_4$ species compared to $C_3$ species (Gowik et al, 2011). Furthermore, two genes in the top-10 have known roles in chlorophyll biosynthesis but no specific link to $C_4$ photosynthesis in literature: GRMZM2G027640 (rank 9), orthologous to the *A. thaliana* light-harvesting-like genes *AT4G17600* and *AT5G47110* (Tanaka et al, 2010), and *ZmELM2* (GRMZM2G101004, rank 10), a heme oxygenase (Shi et al, 2013). In total, seven of the top-10 genes are known to be chloroplast-localized (GRMZM2G111216, GRMZM2G165655, GRMZM2G074393, GRMZM2G543629, GRMZM2G027640, GRMZM2G101004) or light-responsive (GRMZM2G158662), increasing the likelihood that they are involved in processes related to $C_4$ photosynthesis.

## Predicting phenotypic traits of individual plants from leaf transcriptome and metabolome data

We investigated to what extent the transcriptome and metabolome data generated on the individual plants can predict individual plant phenotypes. First, we performed spatially corrected correlation analyses (see Materials and Methods) to identify transcripts and metabolites that show a significant linear association with a given phenotype (Datasets EV11 and EV12). Some of the most interesting transcript–phenotype correlations are briefly discussed below, with homolog or ortholog information derived from the PLAZA database v:4.5 (Van Bel et al, 2018). The interpretation of significantly correlated metabolites is less straightforward however, as most metabolites with significant phenotype correlations have not been identified.

41 genes and 161 metabolites exhibit an expression profile that is significantly correlated ($q \leq 0.05$) with leaf 16 blade length. Notably, the set of significantly negatively correlated genes contains six known or suspected flower development genes: GRMZM2G103666 (*ZmZCN12*) and GRMZM2G051338 (*ZmZCN15*), both phosphatidylethanolamine-binding proteins orthologous to *FLOWERING LOCUS T* (*FT*) in *A. thaliana*; GRMZM2G032339 (*ZmAGL8*), an Agamous-like MADS-box gene; GRMZM2G148693 (*ZmZAP1*) and GRMZM2G553379 (*ZmZMM15*), both MADS-box genes homologous to the *A. thaliana* gene *APETALA1*; and GRMZM2G116658 (*ZmOCL3*), a *HD-ZipIV* homeodomain gene preferentially expressed in the epidermis of reproductive structures and to a lesser extent leaves (Javelle et al, 2011). All of these genes except *ZmOCL3* are in the top-10 of genes most correlated to leaf 16 blade length. The top correlated gene, *ZmZAP1*, was previously found in QTL and GWA studies as a candidate gene associated with ear length (Xue et al, 2016), ear height (Vanous et al, 2018), tassel length (Wang et al, 2018), and flowering time (Wallace et al, 2016), and it has been implicated in maize domestication, in particular for temperate maize lines, in which its expression is downregulated (Liu et al, 2015a).

274 transcript and 133 metabolite profiles are significantly correlated with leaf 16 blade width. Notably, the set of significantly negatively correlated genes again contains *ZmZCN15*, *ZmAGL8*, *ZmZAP1*, *ZmZMM15*, and *ZmOCL3*, and two other known or suspected leaf and flower development genes: GRMZM2G118063 (*ZmHDZIV10*), a *HD-ZipIV* homeodomain gene homologous to *ZmOCL3* (Javelle et al, 2011), and GRMZM2G019317, a LRR

receptor-like kinase orthologous to *SOMATIC EMBRYOGENESIS RECEPTOR-LIKE KINASEs* (*SERKs*) in *A. thaliana*.

583 genes and 241 metabolites have an expression profile that correlates significantly with husk leaf length. The set of genes whose expression in mature leaf 16 tissue negatively correlates with husk leaf length ($q \leq 0.05$, $R^2 > 0.2$) is enriched in leaf and flower development genes and defense response genes ($q \leq 0.05$, Dataset EV11). Next to the flowering genes *ZmZCN15*, *ZmAGL8*, *ZmZAP1*, *ZmZMM15*, *ZmOCL3*, *ZmHDZIV10*, and GRMZM2G019317 that are also found to correlate with leaf 16 blade length or width, the set of putative flowering genes negatively correlated with husk leaf length contains two auxin response factors, GRMZM2G475882 (*ZmARF8*) and GRMZM2G116557 (*ZmARF2*) and one additional MADS-box gene, GRMZM2G059102 (*ZmZMM20*). The set of genes that positively correlate to husk leaf length ($q \leq 0.05$, $R^2 > 0.2$) is enriched in genes involved in, e.g., the response to oxidative stress, salt stress, and UV stress, and also contains the MADS-box gene GRMZM2G171365 (*ZmSOC1*) that was identified in the spatial autocorrelation analysis above as part of a gene cluster correlated with ear length.

118 genes and 74 metabolites exhibit an expression profile in mature leaf 16 tissue that is significantly correlated with ear length at $q \leq 0.05$. No significant GO biological process enrichments were found among positively or negatively correlated genes, but *ZmSOC1* is also identified in this analysis as positively correlated with ear length (Dataset EV11). Interestingly, none of the other flowering genes identified as negatively correlated with leaf 16 blade length, width, or husk leaf length is significantly associated with ear length.

Latsly, 84 genes and 76 metabolites exhibited an expression profile in leaf 16 that is significantly correlated with plant height at $q \leq 0.05$. No significant GO enrichments were found, but interestingly, three of the top-5 of most correlated genes code for transcription factors, among which the photoperiodically regulated transcription factor GRMZM2G107101 (*ZmGI1*, *GIGANTEA1*). *ZmGI1* mutants were found to exhibit early vegetative phase change and early flowering phenotypes under field conditions, and to grow taller than non-mutant plants (Bendix *et al*, 2013). Fittingly, GRMZM2G107101 expression is negatively correlated with plant height in our dataset (Dataset EV11).

The phenotypes of the individual plants can be predicted by the expression patterns of single genes in the leaf 16 blade with maximum $R^2$ scores ranging from 0.407 (for husk leaf length) to 0.292 (for plant height and blade width, Dataset EV11). We investigated whether combinations of genes or metabolites, or both, could lead to a better prediction performance. Elastic net and random forest techniques were used to construct models predicting the phenotypes of individual plants as a function of the transcript and metabolite levels in the harvested leaf samples (see Materials and Methods). Elastic net (e-net) regression is a shrinkage method that is generally well suited for use on high-dimensional datasets (Zou & Hastie, 2005). Its combination of the L1 and L2 penalties of its relatives lasso and ridge regression, respectively, makes e-net regression capable of selecting groups of correlated features (transcripts, metabolites) as predictors. Rather than selecting one representative feature from each group (as in lasso regression), e-nets can select multiple correlated features (as in ridge regression) while still setting the regression coefficients of irrelevant features to zero. This makes the resulting models more biologically interpretable. Random

forest regression (Breiman, 2001) was used in addition because this technique can account for some types of interaction effects between features and is fairly robust to overfitting.

Both types of models were learned for each phenotype using either the transcript levels, the metabolite levels, or both as features (see Table 2 and Datasets EV13-EV16), each time using a 10-fold nested cross-validation strategy (see Materials and Methods). Transcript-based models were additionally run with only a predefined selection of regulatory transcripts as features (see Materials and Methods). The performance of the models was evaluated in two ways: by pooling the predictions for the test sets in each of the 10 folds into one dataset and computing the combined "out-of-bag" (oob) $R^2$ (pooled $R^2$), and by computing the oob $R^2$ on each test fold individually and taking the median (median $R^2$, see Materials and Methods). For all models, 500 datasets with permuted phenotype data were used to compute an empirical *P*-value that reflects whether the $R^2$ score of the model is significantly higher than the $R^2$ scores of models learned on randomized data (see Materials and Methods and Table 2).

Based on the oob $R^2$ scores, blade width and husk leaf length are the phenotypes that are best predictable from the transcriptome and metabolome data, followed by blade length (Table 2 and Fig 7A–F). The transcriptome- and metabolome-based e-net models for leaf 16 blade width reached pooled $R^2$ scores of 0.490 and 0.648, respectively, whereas the $R^2$ values for the best-correlated single gene and metabolite are only 0.292 (Dataset EV11) and 0.350 (Dataset EV12), respectively. This indicates that the multi-feature models for blade width perform substantially better than single-feature models. The performance difference is likely even higher than suggested by the $R^2$ difference, as single-feature models have an advantage in this comparison: multi-gene model $R^2$ values are based on test data while single-gene model $R^2$ values are based on training data. For husk leaf length and blade length, however, the multi-feature models yield $R^2$ scores that are merely comparable to those of the best single-feature models, indicating that only a few genes genuinely contribute to model performance, while inclusion of others in the models leads to data overfitting. The limited data available for model training versus the large number of model features are definitely a factor here (60 and 50 datapoints versus 18,171 and 592 features for transcriptome- and metabolome-based models, respectively).

Ear length and in particular plant height are considerably less predictable than the leaf phenotypes (Table 2 and Fig 7G and H). While all of the models for ear length still reach significant oob $R^2$ scores, most of the models for plant height perform no better than random, and in most cases negative $R^2$ scores are obtained. Tellingly, the multi-feature model oob $R^2$ scores for plant height are much lower than the best single-feature (gene or metabolite) $R^2$ scores (Datasets EV11 and EV12), suggesting that the multi-feature models severely overfit the training data.

These results suggest that predicting phenotypes at the time of sampling gets more difficult as phenotypes become more distantly related to the sampled material (see also Discussion). That the leaf 16-related phenotypes can be predicted better than ear length or plant height is not surprising, as the transcriptome and metabolome data were generated on mature leaf 16 blade tissue. Similarly, husk leaf length is more closely related to the profiled plant material in terms of tissue type than ear length or plant height, which may help explain why it is better predictable.

**Table 2. Performance of e-net and random forest models for trait prediction**

| Trait | | Transcripts | Regulators | Metabolites | Both |
|---|---|---|---|---|---|
| **Pooled $R^2$** | | | | | |
| Blade 16 length | Elastic Net | 0.315 (0.002) | 0.313 (0.002) | 0.359 (0.002) | 0.305 (0.002) |
| | Random Forest | 0.314 (0.002) | 0.421 (0.002) | 0.308 (0.002) | 0.323 (0.002) |
| Blade 16 width | Elastic Net | 0.490 (0.002) | 0.459 (0.002) | 0.648 (0.002) | 0.642 (0.002) |
| | Random Forest | 0.334 (0.002) | 0.272 (0.002) | 0.496 (0.002) | 0.419 (0.002) |
| Husk Leaf length | Elastic Net | 0.458 (0.002) | 0.415 (0.002) | 0.509 (0.002) | 0.476 (0.002) |
| | Random Forest | 0.258 (0.002) | 0.340 (0.002) | 0.555 (0.002) | 0.450 (0.002) |
| Ear length | Elastic Net | 0.235 (0.002) | 0.126 (0.006) | 0.208 (0.002) | 0.279 (0.002) |
| | Random Forest | 0.100 (0.006) | 0.065 (0.018) | 0.287 (0.002) | 0.131 (0.004) |
| Plant height | Elastic Net | 0.058 (0.010) | 0.026 (0.022) | −0.086 (0.681) | −0.048 (0.327) |
| | Random Forest | −0.015 (0.146) | 0.039 (0.030) | 0.018 (0.062) | −0.012 (0.116) |
| **Median $R^2$** | | | | | |
| Blade 16 length | Elastic Net | 0.243 (0.002) | 0.322 (0.002) | 0.079 (0.002) | −0.060 (0.088) |
| | Random Forest | 0.381 (0.002) | 0.473 (0.002) | 0.031 (0.012) | 0.017 (0.016) |
| Blade 16 width | Elastic Net | 0.408 (0.002) | 0.232 (0.002) | 0.668 (0.002) | 0.537 (0.002) |
| | Random Forest | 0.270 (0.002) | 0.321 (0.002) | 0.434 (0.002) | 0.292 (0.002) |
| Husk leaf length | Elastic Net | 0.404 (0.002) | 0.435 (0.002) | 0.449 (0.002) | 0.443 (0.002) |
| | Random Forest | 0.198 (0.002) | 0.233 (0.002) | 0.463 (0.002) | 0.446 (0.002) |
| Ear length | Elastic Net | 0.376 (0.002) | 0.085 (0.002) | 0.370 (0.002) | 0.187 (0.004) |
| | Random Forest | 0.009 (0.016) | −0.051 (0.044) | 0.312 (0.002) | 0.123 (0.002) |
| Plant height | Elastic Net | −0.037 (0.048) | −0.291 (0.824) | −0.166 (0.399) | −0.106 (0.206) |
| | Random Forest | −0.391 (0.858) | −0.323 (0.762) | −0.249 (0.467) | −0.081 (0.074) |
| **PCC** | | | | | |
| Blade 16 length | Elastic Net | 0.587 (0.002) | 0.600 (0.002) | 0.603 (0.002) | 0.562 (0.002) |
| | Random Forest | 0.606 (0.002) | 0.665 (0.002) | 0.566 (0.002) | 0.603 (0.002) |
| Blade 16 width | Elastic Net | 0.717 (0.002) | 0.700 (0.002) | 0.805 (0.002) | 0.812 (0.002) |
| | Random Forest | 0.647 (0.002) | 0.559 (0.002) | 0.753 (0.002) | 0.771 (0.002) |
| Husk leaf length | Elastic Net | 0.677 (0.002) | 0.644 (0.002) | 0.718 (0.002) | 0.690 (0.002) |
| | Random Forest | 0.568 (0.002) | 0.625 (0.002) | 0.757 (0.002) | 0.730 (0.002) |
| Ear length | Elastic Net | 0.503 (0.002) | 0.385 (0.004) | 0.463 (0.004) | 0.535 (0.002) |
| | Random Forest | 0.317 (0.006) | 0.265 (0.016) | 0.542 (0.002) | 0.372 (0.004) |
| Plant height | Elastic Net | 0.268 (0.014) | 0.250 (0.012) | −0.016 (0.122) | −0.135 (0.224) |
| | Random Forest | 0.115 (0.106) | 0.212 (0.030) | 0.172 (0.060) | 0.110 (0.098) |

Three different sections of the table show the pooled $R^2$, median $R^2$, and Pearson correlation (PCC) measures for the prediction performance of the models learned for all traits using all transcripts (Transcripts), only regulatory transcripts (Regulators), all metabolites (Metabolites), and both transcripts and metabolites (Both) as features. Numbers between parentheses indicate $P$-values for the performance values obtained, derived from permutation tests.

For leaf 16 blade width and husk leaf length, the models learned on metabolome data outperform those learned on transcriptome data, despite the fact that they were trained on less data (see Table 2 and Materials and Methods). On the other hand, the feature space of the transcriptome models is much larger than that of the metabolome models (18,171 transcripts versus 592 metabolites), which increases the risk of overfitting for the transcriptome models and may lead to reduced oob $R^2$ scores. Furthermore, for ear length both types of models perform similarly, while for blade length the metabolome models exhibit much lower median $R^2$ values than the

transcriptome models. It is in other words difficult to establish whether transcripts or metabolites are better predictors. Surprisingly, the models learned on both data sources combined did not outperform the models learned on the transcriptome or metabolome data separately. This suggests that most of the relevant phenotype information is redundantly present in both data types. Interestingly, the models learned on the transcriptome data using only the transcript levels of regulatory genes as features performed generally on par with the overall transcriptome models and in terms of median $R^2$ values most often slightly better (Table 2). This indicates that

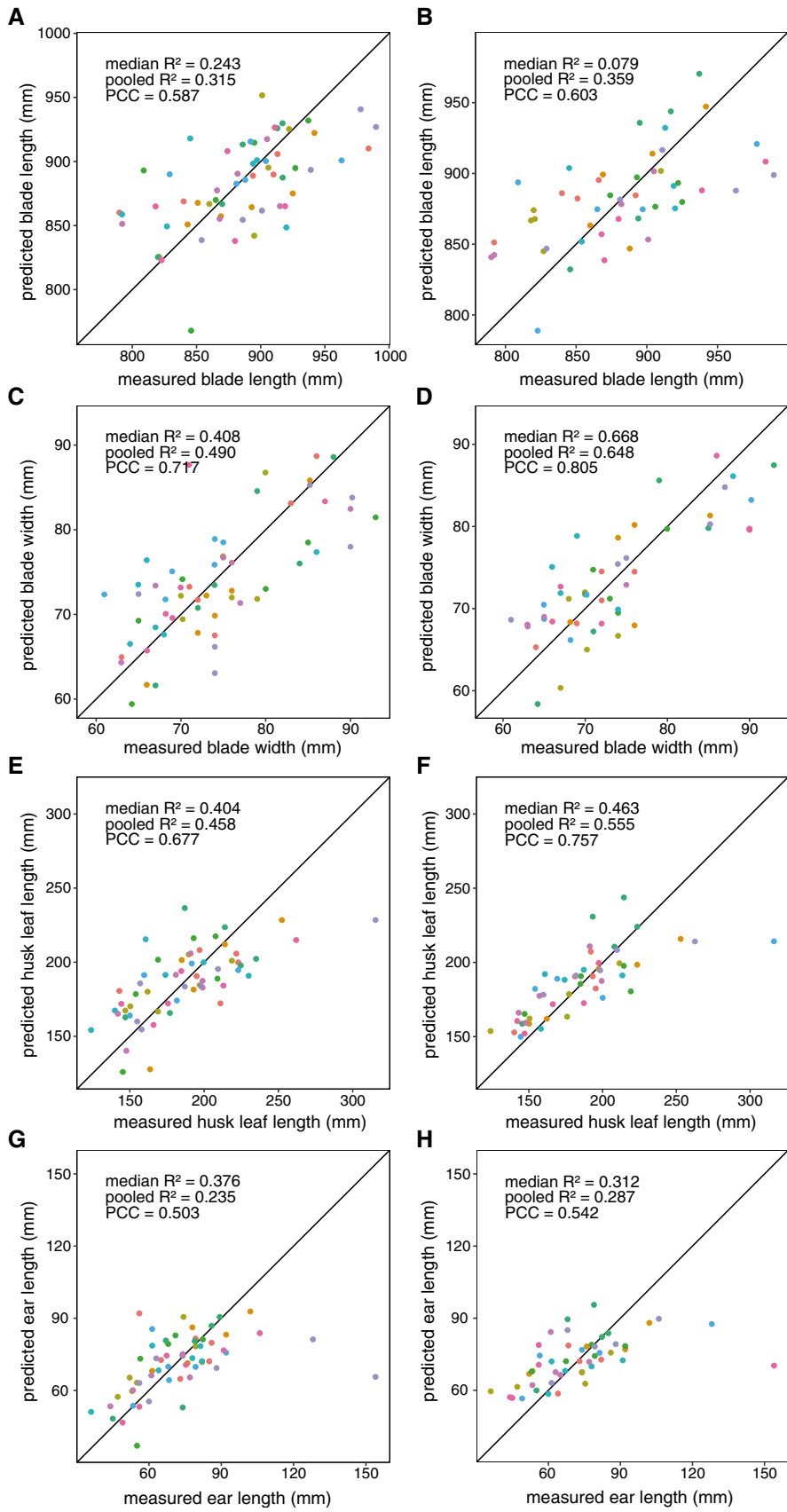

**Figure 7.**

**Figure 7.  Predictive models for leaf 16 blade length and width, husk leaf length and ear length.**

A–H   Graphs plotting predicted versus measured phenotypes are shown for the best-performing whole-transcriptome and metabolome models for each phenotype, based on the pooled $R^2$ scores in Table 2. (A) Transcriptome e-net model for leaf 16 blade length, (B) metabolome e-net model for leaf 16 blade length, (C) transcriptome e-net model for leaf 16 blade width, (D) metabolome e-net model for leaf 16 blade width, (E) transcriptome e-net model for husk leaf length, (F) metabolome random forest model for husk leaf length, (G) transcriptome e-net model for ear length, (H) metabolome random forest model for ear length. The dot colors represent different outer cross-validation folds. Perfect predictions are located on the diagonal line in each panel.

using the expression levels of regulatory genes as features may be sufficient to obtain adequate phenotype predictors, with the advantage that the predictors obtained may be more interpretable from a mechanistic perspective.

We took a closer look at the regulator-based models for leaf 16 blade length and width and for husk leaf length (Datasets EV13-EV15). We focused on the random forest models, as the mean decrease in impurity (MDI) values derived from these models are more interpretable as feature importance statistics than the regularized regression coefficients produced by e-net models. For blade length, two genes have a median MDI score above 0.05 (Dataset EV13): GRMZM2G051338 (*ZmZCN15*, median MDI = 0.234) and GRMZM2G148693 (*ZmZAP1*, median MDI = 0.158). Both genes are found in the top-3 of genes that are most significantly anticorrelated with blade length (Dataset EV11), and both are related to flowering (see above). The next genes in the list have substantially lower median MDI scores, and only 5 genes have a median MDI score above 0.01.

Also for blade width, two genes have a median MDI score above 0.05, but the scores are notably lower than for blade length and more genes (10) have a median MDI score > 0.01 (Dataset EV14). The top-2 blade width regulators with median MDI > 0.05 are GRMZM2G109987 (*ZmRLD1*, *ROLLED LEAF1*, median MDI = 0.0536) and GRMZM2G148693 (*ZmZAP1*, median MDI = 0.0511). *ZmRLD1* codes for a homeobox-leucine zipper transcription factor involved in establishing abaxial-adaxial leaf polarity (Nelson *et al*, 2002). In the semi-dominant *Rld1* mutant, abaxial-adaxial leaf polarity is partially reversed and the leaf blade is transversally rolled inward (Nelson *et al*, 2002). Interestingly, the regulator with the highest elastic net importance, GRMZM2G023625, also has a link to leaf curling. GRMZM2G023625 is a putative HIRA histone chaperone, whose only *A. thaliana* homolog AT3G44530 (*AtHIRA*) is known to be involved in knox gene silencing during leaf development. Reduced *HIRA* expression levels in *A. thaliana* give rise to transversally curled leaves with shorter petioles and often lobes in the proximal region of the blade (Phelps-Durr *et al*, 2005).

For husk leaf length, again two regulatory genes have a median MDI score > 0.05, and 12 genes have a score > 0.01 (Dataset EV15). The top-2 genes are GRMZM2G475014 (*ZmNACTF50*) and GRMZM2G051338 (*ZmZCN15*). *ZmNACTF50* encodes a NAC (No Apical Meristem) transcription factor and is orthologous to AT2G43000 (*JUNGBRUNNEN 1, AtJUB1*) in *A. thaliana*, whose overexpression is known to delay leaf senescence and enhance abiotic stress tolerance (Wu *et al*, 2012). *ZmNACTF50* expression is positively correlated with husk leaf length.

### SNPs in the RNA-seq data have no predictive power for individual plant phenotypes

After establishing that the variability in transcript and metabolite levels among individual maize plants can be used to predict gene

functions and individual plant phenotypes, the question remains to what extent this variability is caused by genetic differences between plants rather than micro-environmental or stochastic differences. Indeed, despite the fact that all plants are from the same inbred line (B104), they still harbor a substantial amount of genetic differences due to somatic and germline mutations and incomplete inbreeding (Appendix Table S6). A GWAS analysis using as features 10,311 biallelic SNPs with minor allele frequency (MAF) ≥ 0.05 after missing data imputation (see Materials and Methods) did not uncover any reliable evidence linking SNPs to phenotypic differences (Appendix Fig S15). A single SNP, which is not associated with a known gene, was found to surpass the significance threshold (Bonferroni-corrected $P ≤ 0.01$) for ear length, but the corresponding quantile–quantile (Q-Q) plot displays abnormalities indicating that this result is likely unreliable (Appendix Fig S15). Given that our dataset contains a low number of samples from the same inbred line and that we can only detect SNPs in the coding and UTR regions of genes from the RNA-seq data, it is far from ideal for GWAS analyses. Nevertheless, the results indicate that the SNPs in the profiled population do not have a major effect on the measured phenotypes.

To confirm this, we also used a subset of 5,007 informative and non-redundant SNPs as features in random forest and e-net models for the phenotypes (see Materials and Methods). None of these models reached a positive oob $R^2$ score (Appendix Table S7), again indicating that the SNPs in the plant population do not significantly influence the phenotypes. Note that the SNPs differentiating the two plant subpopulations are not expected to feature in these models, as the effects of the SNPs concerned have already been removed from the phenotype data during preprocessing. However, the LME models used for SNP effect removal indicate that also this set of SNPs does not significantly influence the measured phenotypes (Appendix Table S1).

## Discussion

In this study, we molecularly and phenotypically profiled 60 individual maize plants of the same inbred line (B104) grown in the same field. Our purpose was to investigate how much information can be extracted from this simple experimental design on the function of genes, and on how gene and metabolite expression relates to plant phenotypes. Although one may expect that this design should yield datasets with a low information content, due to the very limited genetic and environmental variability employed, substantial variability was found in the transcriptomes, metabolomes, and phenotypes of the individual plants. Genes involved in processes such as photosynthesis and stress responses were found to be more variably expressed across the field than housekeeping genes involved in, e.g., RNA and protein metabolism, and the expression patterns of 14.2% of the transcripts and 8.11% of the metabolites profiled

exhibited significant spatial patterning, indicating that the variability uncovered is not merely random noise.

We used the single-plant dataset to predict the function of maize genes from the function of their co-expression network neighbors ("guilt-by-association"), and found that field-grown single-plant transcriptomes overall have higher gene function prediction power than traditional transcriptome datasets profiling pooled plant responses to controlled perturbations in a laboratory. We also found that the single-plant dataset has higher function prediction power than transcriptome data generated on a maize diversity panel (Kremling *et al*, 2018). Furthermore, the single-plant dataset was found to outperform the controlled perturbation datasets for several processes that were likely variably active in the field setting used, in particular abiotic stress responses. This suggests that datasets in which processes are perturbed more subtly around a common baseline may hold an advantage for unraveling gene functions. One of the issues with harsher perturbations is that their effects may propagate further in the cellular networks, and essentially swamp more subtle variations in other, sideways associated processes, decreasing the information content of the resulting data. Pooling samples, although enhancing experimental reproducibility, may similarly decrease the data information content by smoothing out subtle variations across samples.

Comparable results were obtained in an earlier study on individual laboratory-grown *A. thaliana* plants (Bhosale *et al*, 2013). One notable difference with the *Arabidopsis* results however is that the maize single-plant dataset performs better at predicting gene functions at higher (less stringent) *q*-value thresholds, whereas it performs worse at lower *q*-value thresholds (using the performance of traditional treatment/control transcriptome datasets as a baseline). The opposite trend was observed in *Arabidopsis* (Bhosale *et al*, 2013). This is because, taking the precision of predictions from the traditional datasets as a baseline for both species, a disproportionately large fraction of the high-confidence predictions emerging from the maize single-plant dataset are not supported by existing maize gene function annotations. The reason for this is unclear. One difference between both studies is that the individual *A. thaliana* plants were grown in the laboratory, as were the pooled samples they were compared to, while in the present study we use field-generated data for the individual maize plants. Laboratory- versus field-based data generation can however not fully explain the observed precision trend differences, as a decreasing relative precision trend is also observed when comparing the performance of our maize single-plant dataset to that of the sampled diversity panel datasets, for which mature leaf samples were also harvested in the field (Fig EV5). What makes the maize single-plant dataset unique however is that it was generated under fully uncontrolled conditions, whereas for all other datasets some form of deliberate control or treatment was applied. All *A. thaliana* datasets were generated under controlled laboratory conditions, which may have influenced the nature of the data, even for the *A. thaliana* single-plant dataset where no differential treatments were performed. Both the maize SRA datasets and the maize diversity panel data on the other hand contain major "treatment" factors (controlled treatments or genetic diversity, respectively), which may again lead to a different kind of data than profiling untreated individual plants under uncontrolled conditions. We therefore speculate that profiling individual plants of the same line under uncontrolled field conditions may lead to

information about gene function that is complementary to the information gathered from traditional controlled experiments. This may help explain why our single-plant dataset produces high-confidence predictions that are less closely aligned with known gene function annotations, as most of these were derived directly or indirectly (inferred by orthology) from controlled experiments. Confirming the potential value of the novel "false-positive" predictions generated by our field dataset, we found indirect evidence in literature in support of more than 45% of the top-10 novel regulator predictions obtained for $C_4$ photosynthesis, the response to chitin and the response to water deprivation.

Our results indicate that profiling individual plants in the field may also be useful to identify genes that influence plant phenotypes under field conditions. We used machine learning models to quantitatively predict phenotypes of individual plants based on leaf gene expression and metabolome data, and found that leaf phenotypes could be predicted reasonably well, in particular the blade width of leaf 16 (max. median oob $R^2$ score = 0.668 for metabolite e-net model, corresponding Pearson correlation (PCC) between predicted and observed values = 0.805, Table 2). This is fairly remarkable given that the models were learned on data for only 50 or 60 plants. Transcript- and metabolite-based prediction models for leaf phenotypes reached PCC scores in the range 0.57–0.72 and 0.57–0.81, respectively. For comparison, a recent study in which maize phenotypes were predicted from genetic marker and transcriptome data for 388 different maize lines reported maximum PCC values of 0.56 to 0.66 between predicted and measured phenotypes when using both genetic markers and transcript levels as features, and maximum PCC values of 0.51 to 0.61 when using only transcript levels as features (Azodi *et al*, 2020). An important difference however is that the Azodi *et al* (2020) study predicted mature plant phenotypes (final plant height, final yield, flowering time) from seedling data, whereas we predicted actively developing phenotypes from contemporarily profiled leaf transcriptome data. Whereas we could generate decent predictive models for phenotypes that were closely related to the plant material that was molecularly profiled (length and width of the ear leaf blade and length of the developing husk leaf), models learned for more distant phenotypes such as ear length and especially plant height at sampling time performed worse. This discrepancy between the Azodi *et al* (2020) study and ours suggests that intermediate phenotypes may be inherently less predictable than final phenotypes, unless the plant material profiled is directly associated with the phenotype under study. Follow-up experiments will be necessary to assess whether individual plant datasets can be used as efficiently as genomic prediction datasets (Azodi *et al*, 2020) for predicting final plant phenotypes from molecular data profiled at an earlier developmental stage.

Together, our results show that profiling individual plants in the field is a promising addition to the toolbox we have at our disposal to study the molecular wiring of plants and relationships between genes and phenotypes, in particular in a field context. More steps will have to be taken however to realize the full potential of this new experimental design. A major bottleneck in all transcriptome profiling-based strategies to associate genes with phenotypes, not only the single-plant setup but also TWAS and classical systems biology strategies, is that the models they produce are correlational rather than causal in nature. A shift to more causal modeling approaches is direly needed, but not straightforward, as causal

inference from the high-dimensional datasets generated by transcriptome profiling, which are frequently observational in nature and contain lots of hidden variables and confounders, is notoriously difficult. Profiling additional data layers in the single-plant setup, such as micro-environmental variables, may further improve modeling performance and enhance causal interpretability.

Up to now, we only profiled a limited amount of plants of one cultivar in one season and field environment. It remains to be seen to what extent the resulting models can be generalized to other cultivars and growth environments. The fact that the single-plant setup only profiles one specific cultivar at a time may be seen as a disadvantage with respect to the classical TWAS setup, in which multiple cultivars are modeled simultaneously. On the other hand, as the phenotypic effects of expression variants often depend on the genetic background (epistasis) and environment in which they are introduced, it might in fact make sense to study the molecular wiring of a trait in a specific cultivar and environment before attempting generalizations to other cultivars or growth environments, in particular for plant species with large pan-genomes such as maize (Gore *et al*, 2009; Hirsch *et al*, 2014; Lu *et al*, 2015). The single-plant setup might, for instance, be used for studying an elite cultivar directly in a target field environment in which yield or stress tolerance improvements are desired.

More generally, the concept of profiling individuals may also be useful for unraveling gene networks and linking genes to phenotypes in other organisms. This is in fact already done regularly for humans, where performing biological repeats on pools of inbred individuals is not possible. For most other multicellular model organisms, the level of individuals appears to have been skipped, and molecular profiling efforts have moved straight from profiling pools of individuals in three replicates to profiling single cells. We therefore advocate a reappraisal of the level of individuals in systems biology studies.

# Materials and Methods

### Field trial setup, sampling, and phenotyping

During the summer of 2015, 560 B104 maize inbred plants were grown under "uncontrolled" field conditions at a site in Zwijnaarde, Belgium (51°00′35.2″N, 3°42′56.5″E) with a sowing density of 133,333 plants per hectare. Plants were sown by hand in ten adjacent rows of 5.6 m length, 75 cm apart, and each containing 56 maize B104 plants. To the North and West of the B104 plants, the commercial hybrid "Ricardino" was sown, while to the East, more B104 plants were grown and to the South, other hybrids and recombinant inbred lines were grown, separated from the B104 plants by a 2.5 m-wide path (Fig 1A).

In total, 200 non-border plants that exhibited a primary ear at leaf 16 were harvested at the VT (tasseling) stage. Since not all plants reached this stage at the same time, plants were harvested on two different dates, 2015-08-25 (164 plants) and 2015-09-02 (36 plants). On each of these days, harvesting and sampling occurred from 10 am until noon. Damaged plants were discarded to avoid outliers in the data. The position in the field was recorded for the harvested plants, and plant height was measured from the plant base to the collar of the top leaf. The primary ear leaf (leaf 16) of

each selected plant was cut off at the ligule. Leaf 16 blade length was measured from the ligule to the tip of the leaf, while leaf 16 blade width was measured in the middle between the ligule and the leaf tip. For molecular data generation, a 10 cm-long part of the leaf was cut from the middle of the leaf 16 blade, the midrib was removed (to avoid detection of exogenous metabolites during untargeted metabolite profiling), and the resulting mature leaf samples were stored in liquid nitrogen on the field. Primary ears were also cut off from the plants, and the length of the ears and husk leaves (from base to tip) was measured on the field.

### RNA-sequencing

Sixty of the 200 leaf samples for individual plants were randomly selected for RNA-sequencing. Total RNA was isolated with the guanidinium thiocyanate–phenol–chloroform extraction method using TRI Reagent (Sigma-Aldrich). Total RNA was sent to GATC Biotech for RNA-sequencing. Library preparation was done using the NEBNext Kit (Illumina). In brief, purified poly(A)-containing mRNA molecules were fragmented, randomly primed strand-specific cDNA was generated, and adapters were ligated. After quality control using an Advanced Analytical Technologies Fragment Analyzer, clusters were generated through amplification using cBOT (Cluster Kit v4, Illumina), followed by sequencing on an Illumina HiSeq2500 with the TruSeq SBS Kit v3 (Illumina). Sequencing was performed in paired-end mode with a read length of 125 bp, in two batches (see Dataset EV1).

The raw RNA-seq data were processed using a custom Galaxy pipeline (Goecks *et al*, 2010) implementing the following steps. First, the fastq files were quality-checked using FastQC (v:0.5.1) (Andrews, 2010). Next, Trimmomatic (v:0.32.1) (Bolger *et al*, 2014) was used to remove adapters, read fragments with average quality below 10 and trimmed reads shorter than 20 base pairs. The trimmed and filtered reads were mapped to the *Zea mays* B73 reference genome AGPv3.31 (ftp://ftp.ensemblgenomes.org/pub/plants/release-31/fasta/zea_mays/dna/) (Schnable *et al*, 2009) using GSNAP v:2013-06-27 (Wu & Nacu, 2010). A k-mer size of 12 was used, the "local novel splicing event" parameter was set to 50,000, and default values were used for the rest of the parameters. The option for splitting the bam files into unique and multiple alignments was activated, and only the uniquely mapping reads were kept for the following analyses. The mapping files were quantified using HTSeq v:0.6.1p1 (Anders *et al*, 2015) with the option "Intersection-strict" and using the *Zea mays* B73 genome annotation build AGPv3.31 (ftp://ftp.ensemblgenomes.org/pub/plants/release-31/gff3/zea_mays/). The resulting raw counts were filtered to only keep genes with at least 5 counts per million in at least 1 sample. Then, raw counts were divided by size factors calculated by DEseq2 (v:1.14.1) (Love *et al*, 2014), resulting in library size-corrected gene expression values for 18,171 genes across 60 plants. Pseudocounts of $0.5\delta$, with $\delta$ the smallest non-zero value in the normalized expression matrix, were added to all gene expression values. The resulting expression matrix was $\log_2$-transformed.

### Metabolome profiling

Fifty of the 60 leaf samples selected for RNA-sequencing were additionally metabolome-profiled. For metabolome analysis, 100 mg of

frozen, grinded mature leaf 16 material for the selected maize plants was sent to Metabolon Inc. (Durham, NC, USA). Sample extracts were prepared using the automated MicroLab STAR® system from Hamilton Company and divided into five fractions. Samples were normalized based on dry weight and further processed and analyzed by Metabolon for untargeted metabolic profiling involving a combination of four independent approaches: two separate reverse phase (RP)/UPLC-MS/MS analyses with positive ion mode electrospray ionization (ESI), RP/UPLC-MS/MS analysis with negative ion mode ESI and HILIC/UPLC-MS/MS analysis with negative ion mode ESI. All methods utilized a Waters ACQUITY ultra-performance liquid chromatographer (UPLC) and a Thermo Scientific Q-Exactive high resolution/accuracy mass spectrometer interfaced with a heated electrospray ionization (HESI-II) source and an Orbitrap mass analyzer operated at a mass resolution of 35,000. Sample extracts were dried and then reconstituted in solvents compatible to each of the four methods. Each reconstitution solvent contained a series of standards at fixed concentrations to ensure injection and chromatographic consistency. One aliquot was analyzed using acidic positive ion conditions, chromatographically optimized for more hydrophilic compounds. In this method, the extract was gradient eluted from a C18 column (Waters UPLC BEH C18-2.1x100 mm, 1.7 μm) using water and methanol, containing 0.05% perfluoropentanoic acid (PFPA) and 0.1% formic acid (FA). Another aliquot was analyzed using acidic positive ion conditions, chromatographically optimized for more hydrophobic compounds. In this method, the extract was gradient eluted from the same aforementioned C18 column using methanol, acetonitrile, water, 0.05% PFPA and 0.01% FA and was operated at an overall higher organic content. Another aliquot was analyzed using basic negative ion optimized conditions using a separate dedicated C18 column. The basic extracts were gradient eluted from the column using methanol and water, however with 6.5 mM Ammonium Bicarbonate at pH 8. The fourth aliquot was analyzed via negative ionization following elution from a HILIC column (Waters UPLC BEH Amide 2.1 × 150 mm, 1.7 μm) using a gradient consisting of water and acetonitrile with 10mM Ammonium Formate, pH 10.8. The MS analyses alternated between MS and data-dependent MS scans using dynamic exclusion. The scan range varied slighted between methods but covered 70–1,000 *m/z*. Raw data was extracted, peak-identified and QC processed using Metabolon's hardware and software. Compounds were identified by comparison to library entries of more than 3,300 purified standards or recurrent unknown entities. Metabolon's library was based on authenticated standards that contain the retention time/index (RI), mass to charge ratio (*m/z*), and chromatographic data (including MS/MS spectral data) of all molecules present in the library.

The metabolite profiles used in the downstream analyses were obtained from the raw data delivered by Metabolon Inc. as follows. Log$_2$ transformation was applied to the initial matrix containing the levels of 601 metabolites across 50 samples. Outliers were identified iteratively using two-tailed Grubbs tests (threshold for outlier detection was $P = 0.01$) and converted to missing values (NA). Metabolites with missing values for more than half of the samples were removed, resulting in a matrix containing the levels of 592 metabolites across 50 samples. To deal with residual missing values, imputation was performed using Bayesian principal component analysis (BPCA) with 48 components (using the pca function of the pcaMethods R package, v:1.76.0 with method= "bpca", scaling= "uv" (unit

variance), npcs = 48). Finally, quantile normalization was applied to give each sample the same data distribution.

## SNP detection and population structure analysis

Aligned reads for variant calling were obtained using HISAT2 v:2.1 (Kim *et al*, 2015) with default parameters. Variants were identified using NGSEP v:3.3.2 (Tello *et al*, 2019). For downstream analyses, we focused on biallelic SNPs with a minimum genotype quality of 40 and called in at least 48 samples (80% of the population). Missing calls were imputed using Beagle v:5.1 (Browning *et al*, 2018) using default parameters, and only SNPs with minor allele frequency (MAF) ≥ 0.05 after imputation were kept, resulting in a dataset of 10,311 SNPs.

A neighbor-joining tree was made based on the SNP dataset with TASSEL v:5.2.60 (Bradbury *et al*, 2007), using 1—IBS (identity by state) as the distance measure while setting the distance from an individual to itself to zero. The tree was rendered using the polar tree layout in FigTree v:1.4.3 (Rambaut, 2016). Principal component analysis (PCA) of the SNP dataset was done with the R package ggfortify v:0.4.10 (Tang *et al*, 2016), using at each locus the genotype encoding 0, 1, and 2 for the homozygous reference genotype, the heterozygous genotype, and the homozygous alternative genotype, respectively. The PCA results were plotted with the R package factoextra v:1.0.7 using the repel option.

## Analysis and correction of systematic effects in the single-plant data

To assess sequencing batch effects, day-of-harvest (DOH) effects, SNP population structure effects, and spatial autocorrelation effects on the transcriptome dataset, the following linear mixed-effects (LME) model was used:

$$\mathbf{y} = \beta_I + \beta_b \mathbf{x}_b + \beta_d \mathbf{x}_d + \beta_s \mathbf{x}_s + \boldsymbol{\varepsilon} \tag{1}$$

Here, $\mathbf{y}$ is a vector of log$_2$ expression levels for a given gene $g$ across samples. $\beta_I$ is the intercept (average gene expression), and $\beta_b$, $\beta_d$, and $\beta_s$ are coefficients for the batch, DOH, and SNP effects, respectively. The vectors $\mathbf{x}_b$, $\mathbf{x}_d$, and $\mathbf{x}_s$ encode the batch, DOH, and SNP groups across the sampled plants, using 0 for the reference group of plants (= biggest group for the effect concerned) and 1 for the alternative group. The model errors $\boldsymbol{\varepsilon}$ follow a multivariate normal (MVN) distribution with a spherical covariance structure:

$$\boldsymbol{\varepsilon} \sim N(0, \textstyle\sum) \text{ where } \sum_{ij} = \sigma^2\left(nI_{ij} + (1-n)\mathrm{corSpher}(i,j)\right) \tag{2}$$

$$\mathrm{corSpher}(i,j) = \begin{cases} d_{ij} < r & 1 - 1.5\dfrac{d_{ij}}{r} + 0.5\left(\dfrac{d_{ij}}{r}\right)^3 \\ \text{otherwise} & 0 \end{cases}$$

where $\sum$ is the covariance matrix, $\sigma$ is the overall magnitude of expression noise for gene $g$ (comparable to the standard deviation of a univariate normal distribution), and $n$ is a nugget parameter bounded between 0 and 1 quantifying the proportion of independent and identically distributed (*i.i.d.*, i.e., not spatially autocorrelated) noise in the expression of gene $g$. The "corSpher" function decreases

from 1 to 0 as the field distance $d_{ij}$ between plants $i$ and $j$ increases. The spherical covariance structure was chosen as it gave the most meaningful range estimates (within bounds of the field when $n \neq 1$), but other covariance structures yield similar results. $r$ is a range parameter related to the distance at which the expression of gene $g$ becomes independent between plants. If $n = 1$ or $r = 0$, the model reduces to a simple linear regression model.

The model was optimized using restricted maximum likelihood estimation (REML) using the *gls* function of the nlme package v:3.1-148 (Pinheiro *et al*, 2019) in R v:4.0.2. The parameters $\beta_I$, $\beta_b$, $\beta_d$, $\beta_s$, $\sigma$, $n$, and $r$ are estimated from the data for each gene. In case of convergence errors (when $n$ approaches 1 or $r$ approaches 0), ordinary least-squares (OLS) regression was used instead.

The same model and estimation procedure were used on the metabolome and phenotype data, except that the sequencing batch effect is not relevant for these datasets and was hence left out of the model. To assess the proportion of variance explained by systematic effects in the transcriptome, metabolome, and phenotype datasets, we estimated $R^2$ values for the LME model effects per gene/metabolite/phenotype as described in Nakagawa and Schielzeth (2013). There is however no consensus on how to estimate $R^2$ in LME models, so these values should be approached with caution. The total proportion of variance explained by the fixed model effects was calculated as:

$$R^2_{fix} = \frac{\text{var}(f)}{\text{var}(f) + \text{var}(\varepsilon)} \qquad (4)$$

where $\text{var}(\varepsilon)$ is the error variance and $\text{var}(f)$ is the variance of all fixed effects combined, i.e., $\text{var}(f) = \text{var}(\beta_b \mathbf{x}_b + \beta_d \mathbf{x}_d + \beta_s \mathbf{x}_s)$. The proportion of variance explained by each fixed effect component separately is calculated similarly, e.g., for the DOH effect:

$$R^2_{DOH} = \frac{\text{var}(\beta_d \mathbf{x}_d)}{\text{var}(f) + \text{var}(\varepsilon)} \qquad (5)$$

Although the batch, DOH, and SNP effects are not significantly correlated (10,000 permutation tests, $P > 0.10$), $\text{var}(\beta_b \mathbf{x}_b + \beta_d \mathbf{x}_d + \beta_s \mathbf{x}_s) = \text{var}(\beta_b \mathbf{x}_b) + \text{var}(\beta_d \mathbf{x}_d) + \text{var}(\beta_s \mathbf{x}_s)$ and hence $R^2_{fix} = R^2_{batch} + R^2_{DOH} + R^2_{SNP}$ only holds approximately due to limited sampling effects.

The proportion of variance contained in the model residuals (errors) is calculated as:

$$R^2_\varepsilon = \frac{\text{var}(\varepsilon)}{\text{var}(f) + \text{var}(\varepsilon)} \qquad (6)$$

As a proportion $(1 - n)$ of the noise is estimated to be spatially autocorrelated and a proportion $n$ is estimated to be *i.i.d.*, $R^2_{cov} = (1 - n)R^2_\varepsilon$ and $R^2_{iid} = nR^2_\varepsilon$ were taken as crude measures for the proportion of variance explained by spatial autocorrelation effects and *i.i.d.* noise, respectively. Again, these $R^2$ values need to be interpreted cautiously.

As we want to assess the use of inter-individual variability among comparable field-grown plants for predicting gene functions and plant phenotypes, between-group variability caused by batch, DOH, and SNP effects was removed from the data before downstream analyses.

## Spatial autocorrelation analyses

Spatially autocorrelated transcripts, metabolites, and phenotypes were detected using Moran's I with an inverse distance-weighted matrix in the Ape package v:5.4 (Paradis & Schliep, 2018) in R v:4.0.2. The *P*-values computed by the Ape package were adjusted for multiple testing with the Benjamini–Hochberg (BH) procedure (Benjamini & Hochberg, 1995), which controls the false discovery rate (FDR). The *z*-scored profiles of transcripts and metabolites with $q \leq 0.01$ were assigned to clusters using hierarchical clustering ("hclust" function in R using "ward.D2" linkage). Associations between a given spatially autocorrelated transcript or metabolite cluster and any phenotypes were assessed by testing for Pearson correlation between the average *z*-scored gene expression profile of the cluster and the phenotype profiles. The resulting *P*-values were corrected per phenotype using the BH method.

## Analysis of gene and metabolite expression variability

Coefficients of variance (CVs) were calculated on non-log transformed transcriptome and metabolome data for the individual plants after correction for sequencing batch, DOH, and SNP effects (see above). To this end, the corrected data were inverse $\log_2$-transformed and, for the transcriptome data, a pseudo-count was subtracted from this back-transformed matrix so that the minimal expression value in the matrix was again zero.

To assess potential bias in metabolite CV depending on the average levels of metabolites, a linear regression trendline was fit for the $\log_{10}(CV^2)$ versus $\log_{10}(\text{mean})$ relationship. To assess the bias in transcript CV depending on the average expression of genes, a trendline was fit for the $CV^2$ versus mean relationship with a generalized linear model (GLM) of the gamma family with identity link of the form $CV^2(\mathbf{x}) = a/\overline{\mathbf{x}} + b$ for gene expression profiles $\mathbf{x}$ with fitting parameters $a$ and $b$, as in Brennecke *et al* (2013). We used code for this based on the "BrenneckeGetVariableGenes" function in the M3Drop R package (Andrews & Hemberg, 2019). The 5% genes with the lowest mean expression were removed from the variance analysis before fitting the transcriptome trendline. To correct for the observed mean–variance relationships, a normalized CV score was computed for each transcript or metabolite profile $\mathbf{x}$ as in Cortijo *et al* (2019): $normCV(\mathbf{x}) = \log_2\left(CV^2(\mathbf{x})/trend(\overline{\mathbf{x}})\right)$, with $trend(\overline{\mathbf{x}})$ the value of the fitted trendline at the mean of $\mathbf{x}$. The top and bottom 10% of genes ranked by decreasing *normCV* score were labeled as highly variable and lowly variable genes, respectively.

For comparison with the expression data of Cortijo *et al* (2019), the raw mapped read counts for two time points in the latter dataset (ZT06 and ZT20) were filtered, normalized, and $\log_2$-transformed as described above for the maize single-plant dataset. Sequencing batch effects were removed for each time point separately as described in Cortijo *et al* (2019), based on spike-ins using the "RUVg" function in the RUVSeq R package (Risso *et al*, 2014) (with the internal RUVSeq log-transform and inverse log-transform functionality disabled). Inverse log-transformation, trendline fitting, and computation of *normCV* scores were done as for the maize single-plant dataset.

## Clustering analyses

The transcriptome and metabolome datasets were $z$-scored and jointly clustered using the ward.D2 hierarchical clustering method (Murtagh & Legendre, 2014) included in the R stats package (v:4.0.2), and using squared Euclidean distance as the distance measure. The same protocol was used for clustering the RNA-seq datasets sampled from the Sequence Read Archive v. 2018/01/30 (https://www.ncbi.nlm.nih.gov/sra) (Leinonen *et al*, 2011) (see further). Additionally, the single-plant transcriptome dataset was analyzed using the biclustering algorithm ENIGMA v:1.1 (Maere *et al*, 2008). For biclustering, the $\log_2$ expression values were transformed to $\log_2$ fold changes with respect to the mean gene expression across the individual plants. Default parameter settings were used, except for "*fdr*" = 0.005, "*fdrBiNGO*" = 0.01, "*namespaces*" = biological_process and "*pvalThreshold*" = 0.5943369. The latter threshold is the standard deviation of the $\log_2$ fold changes across the entire RNA-seq dataset, which, by lack of differential expression $P$-values for the single plants, is used by ENIGMA as a threshold for discretizing transcript $\log_2$ fold changes into the categories "upregulated", "downregulated" and "unchanged".

## Gene Ontology (GO) enrichment analyses

The gene ontology file used for GO enrichment analyses was downloaded from the Gene Ontology knowledgebase (http://www.geneontology.org) (The Gene Ontology Consortium, 2017). A GO annotation file for maize B73 AGPv3.31 genes was parsed from the functional annotations provided by PLAZA (Proost *et al*, 2015), development version cnb 02, on November 27, 2017. To ensure that all the functional annotations found for the genes in the AGPv2 maize genome were included in our analyses, we also included the maize gene functional annotations provided by the older PLAZA 3.0 platform (Proost *et al*, 2015), taking into account gene identifier changes from AGPv2 to AGPv3 as recorded in MaizeGDB (https://www.maizegdb.org) (Portwood *et al*, 2018). Given the lack of maize genes annotated to the $C_4$ photosynthesis category in GO, we manually added annotations to this category for 78 genes identified as $C_4$ genes by Li *et al* (2010). In all GO enrichment analyses, enrichment $P$-values were calculated using hypergeometric tests and adjusted for multiple testing ($q$-values) using the Benjamini–Hochberg (BH) procedure (Benjamini & Hochberg, 1995). For GO enrichment analyses on (bi)clustering results, multiple testing correction was done for each cluster separately. Genes annotated to the categories "*DNA binding transcription factor activity*" (GO:0003700), "*signal transducer activity*" (GO:0004871), and "*regulation of transcription - DNA-templated*" (GO:0006355) were combined in a list of potential regulators (Dataset EV17), for use in the ENIGMA analysis, the literature screen for evidence supporting our gene function predictions, and some of the phenotype prediction models, namely those that use a predefined list of regulators as potential predictors (see further).

## Correlation network generation for gene function prediction

For each pair of genes $x$ and $y$ in the single-plant transcriptome dataset, a "spatially adjusted Pearson correlation" was computed by $z$-scoring the $\log_2$ gene expression profiles of both genes and fitting the following model to the data:

$$\mathbf{y} = \beta \mathbf{x} + \boldsymbol{\varepsilon} \tag{7}$$

with $\beta$ the correlation coefficient and $\boldsymbol{\varepsilon}$ an error term with a spherical covariance structure as in Equation 2. Model parameters ($\beta$ and the spherical covariance parameters $r$, $n$, $\sigma$) were optimized with restricted maximum likelihood optimization (REML) using the *gls* function of the nlme package v:3.1-148 (Pinheiro *et al*, 2019) in R v:4.0.2. Although there is an asymmetry in the regression equation, swapping $x$ and $y$ for gene pairs with a range estimate $r$ above zero gave parameter estimates that were not meaningfully different.

For most gene pairs $r$ converged to zero or $n$ converged to 1, which means the best-fit model is one without spatial covariance, yielding the exact same correlation coefficient $\beta$ and corresponding $P$-value as a normal ordinary least-squares (OLS) regression or Pearson correlation on the $z$-scored variables (up to rounding errors). For about 23% of the gene pairs, $r$ converged to a non-zero distance. This means that for these gene pairs, there would be spatial structure left in the residuals of an OLS regression, violating the assumption of independence in OLS regression. All $P$-values were Bonferroni-corrected, and correlations with corrected $P$-values ≤ 0.01 were included as edges in the correlation network.

The expression correlation network obtained from the single-plant dataset was compared with networks obtained from traditional RNA-seq datasets sampled from the Sequence Read Archive (https://www.ncbi.nlm.nih.gov/sra) (Leinonen *et al*, 2011). The raw RNA-seq data downloaded from the SRA in first instance involved all transcriptome data on *Zea mays* profiled with Illumina sequencing platforms. Only runs profiling mRNA (as opposed to, e.g., small RNAs) with an average read length > 30 bp and $\geq 4.10^6$ reads were retained. In many cases, the meta-information obtained from SRA did not specify the genotype and tissue profiled in the RNA-seq experiments. We therefore used information from the BioSample database (https://www.ebi.ac.uk/biosamples/) to select only RNA-seq datasets produced on leaves of the maize inbred line B73, discarding crosses, mutants, and NILs. Only samples with a unique BioSample ID were retained to avoid data replication. This led to a compendium of 470 unique RNA-seq samples (Dataset EV8), which were preprocessed and normalized in the same way as the single-plant samples. As an additional data quality filtering step, samples with < 80% uniquely mapping reads, samples with a clearly divergent data distribution and samples with less than 20,000 expressed genes were discarded. This resulted in a compendium of 407 RNA-seq samples, which we randomly sampled without replacement to extract 500 compendia of 60 samples. For each of these randomly sampled compendia, a correlation network was built using Pearson correlation. Note that in contrast to the single-plant dataset, spatial autocorrelation correction is not necessary for the datasets sampled from SRA. Every sampled network was thresholded to obtain the same number of edges as obtained for the single-plant network.

The single-plant network was additionally compared with networks inferred from expression data on a maize diversity panel (Kremling *et al*, 2018). To maximize the comparability of the diversity and single-plant networks in terms of the tissue and time of day profiled, we focused on 210 Leaf Mature Adult Day (LMAD) samples in the Kremling *et al* (2018) dataset. The diversity samples were preprocessed and normalized in the same way as the single-plant samples, and 100 compendia of 60 samples were randomly sampled

without replacement from the diversity dataset. A Pearson correlation network was built for each diversity compendium and thresholded to obtain the same number of edges as in the single-plant network.

## Gene function prediction

Gene functions (GO biological process annotations) were predicted from the single-plant correlation network, the 500 sampled SRA networks, and the 100 sampled diversity networks using a command-line version of PiNGO (v:1.11) (Smoot *et al*, 2011). PiNGO predicts the function of a given gene based on the GO annotations of its neighbors in a given network, using hypergeometric GO enrichment tests on the gene's network neighborhood. The resulting *P*-values were adjusted for multiple testing (for each input network separately) using the BH method. The overall function prediction performance of the single-plant and sampled networks was calculated as in Bhosale *et al* (2013). Recall and precision of the functional predictions for a given gene in a given network were calculated as described by Deng *et al* (2004) using the known maize GO annotations as gold standard, and the overall recall and precision values for the given network were obtained by averaging across all genes in the network. Next to this overall analysis of gene function prediction performance, we also assessed how accurately the SRA networks and the single-plant network predicted genes involved in specific GO biological processes. For these analyses, recall ($R$) and precision ($P$) were calculated in the traditional way as $R = tp/(tp+fn)$ and $P = tp/(tp+fp)$ with $tp$ the number of true positives, $fp$ the number of false positives, and $fn$ the number of false negatives identified.

For every GO category and overall, the recall, precision, and *F*-measure (harmonic mean of recall and precision) of the predictions were calculated for every network at prediction *q*-value thresholds ranging from $10^{-2}$ to $10^{-11}$. Undefined precisions and F-measures, resulting from a network not producing any predictions at a given *q*-value threshold, were set to 0 in order to reflect poor performance of the network at the *q*-value concerned. The relative prediction performance of the single-plant network with respect to the sampled SRA networks was classified as very good, good, average, poor, or very poor as in Bhosale *et al* (2013), based on the root mean square deviation of the single-plant network *F*-measures from the 25th, 50th, and 75th percentiles of the sampled network *F*-measures over the FDR subrange in which either the single-plant network or at least 250 of the 500 sampled networks, or both, exhibited non-zero *F*-measures.

## Predictive models for phenotypes

Phenotypes were regressed on the expression of single genes and metabolites using a mixed model with the following formulation:

$$\mathbf{y} = \beta_0 + \beta \mathbf{x} + \boldsymbol{\varepsilon} \tag{8}$$

with $\mathbf{x}$ the $\log_2$ expression profile of a given gene or metabolite and $\mathbf{y}$ the corresponding vector of phenotype values across plants. The error $\boldsymbol{\varepsilon}$ is assumed to follow a multivariate normal distribution with a rational quadratic distance-based covariance function. That is, the covariance of $\boldsymbol{\varepsilon}$ is described by:

$$\mathrm{cov}(i,j) = \sigma^2 \times (n + (1-n) \times \mathrm{corRatio}(i,j)) \tag{9}$$

where $\sigma$ is the magnitude of the noise and the nugget $n$ determines which proportion of the residuals is governed by spatial auto-covariance. The correlation function $\mathrm{corRatio}(i,j)$ between two samples $i$ and $j$ is given by:

$$\mathrm{corRatio}(i,j) = 1 / \left(1 + (d_{ij}/r)^2\right) \tag{10}$$

with $d_{ij}$ the physical distance between plants $i$ an $j$ in the field. The range parameter $r$ is related to the distance at which two plants become independent of one another. The ratio kernel was chosen because it gave meaningful range estimates (Appendix Fig S16) and the best overall performance as measured by BIC. Regression analyses were performed using the nlme package (v:3.1-148) (Pinheiro *et al*, 2019) in R (v:4.0.2). *P*-values were adjusted for each phenotype separately using the BH method.

Elastic net (e-net) and random forest methods were used to learn multi-feature predictive models for the phenotypes using the *z*-scored $\log_2$ transcript levels, metabolite levels or both as features. E-net and random forest models were also built using as features only the transcript levels of a predefined set of regulators (Dataset EV17) and using as features a set of 5,007 SNPs. These were derived from the previously identified set of 10,311 biallelic SNPs by removing 2,246 SNPs that were heterozygous for all plants (and thus uninformative) and collapsing 939 groups of perfectly correlated SNP profiles (involving 3,997 SNPs in total) into 1 representative profile per group.

E-net and random forest models were built with the scikit-learn package (v:0.21.0) (Pedregosa *et al*, 2011) in Python. For e-net models, the maximum number of iterations (parameter "*max_iter*") was set to $10^6$. For random forest models, the number of estimators, i.e., the number of averaged trees, was set to 500, the "*criterion*" parameter was set to "mse", and the "*bootstrap*" parameter was set to "True". For each phenotype, models were built with each method on each feature set using 10-fold nested cross-validation. For each of the 10 outer folds, 4 inner folds were used to tune the model hyperparameters (the shrinkage parameter $\alpha$ and the L1-ratio $\rho$ for elastic nets; the "*max_features*" parameter with possible values "sqrt", 0.33, "log2" and "None" and the "*min_samples_split*" parameter with possible value 2, 3, 4, and 5 for random forests). After completing the inner cross-validation, the combination of hyperparameters that scored best on test data across the 4 folds was used to retrain the model on all 4 folds combined, yielding 10 trained models with optimized hyperparameters per phenotype (GridSearchCV function in scikit-learn). Each of the 10 models was used to predict the phenotypes of the hold-out samples (6 per fold for transcripts, 5 for metabolites) for the fold it was trained on, yielding 60 transcriptome-based or 50 metabolome-based "test data" predictions in total, one for each sample.

The "out-of-bag" (oob) $R^2$ score, defined as $R^2 = 1 - \sum(y_i - \hat{y}_i)^2 / \sum(y_i - \bar{y})^2$ where $\hat{y}_i$ and $y_i$ are the predicted and observed phenotypes for sample $i$, respectively, and where $\bar{y}$ is the mean of the observed phenotypes, was used to measure how well the predictions align with the true phenotypes. Note that the meaning of this oob $R^2$ is different from the classical meaning of $R^2$, which is the percentage of variance explained by a linear model. As opposed to

the classical $R^2$, the oob $R^2$ can become negative when the sum of squared errors (numerator) is larger than the variance of the data (denominator). When all predictions $y$ equal the mean $\bar{y}$, the oob $R^2$ equals zero. A negative oob $R^2$ score[i] indicates that the model does worse than assigning the mean phenotype value of the test samples to all test samples. Positive oob $R^2$ scores indicate that the model does better than predicting the mean, and a model that perfectly predicts the unseen phenotypes has an oob $R^2$ score of one. We report two oob $R^2$ scores for each model, the "pooled $R^2$" score and the "median $R^2$" score. For calculating the pooled $R^2$, the test set predictions of all folds were taken into account together to calculate one oob $R^2$ value that summarizes all folds. The "median $R^2$" score is the median of the oob $R^2$ scores calculated for each fold independently.

For modeling methods that use built-in feature selection/reduction techniques, such as e-nets and random forests, an analytical statistical framework to assess whether models perform better than expected by chance is lacking. A typical solution used is to compute empirical *P*-values by applying the same data analysis to a large number of datasets that follow the null hypothesis of no relation between the dependent and independent variables, and comparing the parameter values and performance measures of the model to their empirical null distributions (Ojala & Garriga, 2010; Riedelsheimer *et al*, 2012; Steinfath *et al*, 2010). 500 datasets following the null hypothesis of no relation between gene or metabolite expression and phenotypes were generated by randomly permuting the phenotypes among the plants. The following formula (Ojala & Garriga, 2010) was used to calculate *P*-values for the original oob $R^2$ scores:

$$p = \frac{n+1}{k+1} \tag{11}$$

where $n$ is the number of times that a permuted model gave an equal or better $R^2$ score than the "true" model and $k$ is the number of permutations. Following Ojala & Garriga (Ojala & Garriga, 2010), the standard deviation on the empirical *P*-value can be calculated as $\sqrt{\frac{P^*(1-P^*)}{k}}$, where *P** is the true *P*-value. This underlying true *P*-value is unknown, but at the critical $P^* = 0.05$, the calculated standard deviation on the empirical *P*-value when using 500 permutations is 0.0097, which is sufficiently low for our purposes.

### GWAS analysis

GWAS analysis was performed in TASSEL v:5.2.60 (Bradbury *et al*, 2007) with mixed linear models (MLMs) encoding population structure as a fixed effect and a kinship matrix as random effect. To account for population structure, the plant coefficients for the first two principal components of the SNP-based PCA were used as covariates (see above). Note that this population structure correction is not strictly necessary as the observed population structure effect was already removed from the phenotype data during preprocessing. The kinship matrix was built within TASSEL using the option "Centered identity by state IBS" with default parameters. Manhattan and Q-Q plots were made using the R package qqman v:0.1.4 (Turner, 2018).

## Data availability

The datasets and computer code produced in this study are available in the following databases:

- RNA-seq data: Dataset EV1 and ArrayExpress E-MTAB-8944 (https://www.ebi.ac.uk/arrayexpress/experiments/E-MTAB-8944).
- Metabolomics data: Dataset EV1 and Zenodo (https://zenodo.org/record/4034433).
- Phenotype data: Dataset EV1 and Zenodo (https://zenodo.org/record/4034433).
- Data analysis scripts: Zenodo (https://zenodo.org/record/4034433).

**Expanded View** for this article is available online.

### Author contributions

SM designed the study. SM, HN, and DI supervised the study. TVH, JDB, HN, DH, and SM performed the field trial and generated data. DFC, SDM, JA, HS, DH, and SM analyzed data. SM, DFC, and SDM wrote the manuscript with input from the other authors.

### Acknowledgements

The authors thank Alex de Vliegher and his team from the Flanders Research Institute for Agriculture, Fisheries and Food (ILVO) for field trial management, members of various laboratories at the VIB-UGent Center for Plant Systems Biology for assistance with harvesting, Karl Kremling for advice on analysis of the diversity panel data, Ethalinda Cannon for greatly facilitating data retrieval from MaizeGDB, and three anonymous reviewers for very helpful comments. Funding for the RNA-seq and metabolomics data generation and funding for the work of DH and TVH were provided by Syngenta Crop Protection, LLC. SDM is a fellow of the Research Foundation-Flanders (FWO, grant 1146319N).

### Conflict of interest

The authors declare that they have no conflict of interest.

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
