## [Review Process File · Molecular Systems Biology]

Using single-plant -omics in the field to link maize genes to functions and phenotypes

Daniel Cruz, Sam De Meyer, Joke Ampe, Heike Sprenger, Dorota Herman, Tom Van Hautegeem, Jolien De Block, Dirk Inzé, Hilde Nelissen, and Steven Maere

DOI: [10.15252/msb.20209667](https://doi.org/10.15252/msb.20209667)

Corresponding author(s): Steven Maere (steven.maere@ugent.vib.be)

Review Timeline:

Submission Date:	28th Apr 20
Editorial Decision:	10th Jun 20
Revision Received:	18th Sep 20
Editorial Decision:	22nd Oct 20
Revision Received:	29th Oct 20
Accepted:	17th Nov 20

Editor: Jingyi Hou

Transaction Report:

Thank you for submitting your work to Molecular Systems Biology. We have now received all the reports and as you will see below, the reviewers think that the presented methodology and findings seem interesting. They raise however a series of concerns, which we would ask you to convincingly address in a revision.

The recommendations provided by the reviewers are very clear and therefore there is no need to reiterate the points listed below. In particular, Reviewer #3 is concerned about the lack of experimental validation of the novel gene functions. If you have such data at hand, we would encourage you to include it in the revised manuscript. However, this is not mandatory for publication. Further, in light of the concerns of Reviewer #1, we would ask you to edit the manuscript to make sure that the main findings are sufficiently clear and easily accessible to the general audience of Molecular Systems Biology.

All other issues raised by the reviewers would need to be convincingly addressed. As you may already know, our editorial policy allows in principle a single round of major revision and it is therefore essential to provide responses to the reviewers' comments that are as complete as possible. Please feel free to contact me in case you would like to discuss in further detail any of the issues raised by the reviewers.

On a more editorial level, please do the following.

Reviewer #1:

In the work of Cruz et al., The authors take an interesting approach, examining the transcriptomes and metabolomes of individually grown maize plants in the field, and comparing these results to the individual phenotypes of the plants. In general, this appears a promising approach to understanding and transferring lab based plant results to the field. It reveals a high level of phenotypic variability between plants in the field which can to some degree be matched to transcriptional differences between plants. In general, I think the paper is an interesting and important step forward for the plant systems biology field. I have the following concerns:

Readability for a general audience: Many of the aspects of this study will be very interesting to the general audience of Molecular Systems Biology. However, it is written for a plant audience currently. It would aid the impact of the work if the introduction and discussion could be a little less plant specific. The issue of individual variability within populations has been explored in microbial and mammalian systems, and I would think that the MSB readers would appreciate a link to this work, as well as more general points to be made about the use of using the variability between individuals under the same environment for generating regulatory networks.

Analysis of Individual transcriptome data. In examining individual transcriptional variability (either single cell or single individual) it is important to correct for technical noise (e.g, <https://doi.org/10.1038/nmeth.2645>). As detected levels get lower, CV increases due to this. Hence, it is important to use, or consider, a corrected CV score to account for this. I can't see a figure

showing how CV scales with the mean expression of genes in the paper. It would be interesting if this could be added, as well as a more detailed explanation for why CV was chosen as the measure of variability. Additionally, the authors write 'The average transcript level CV of ~0.3 is about three times higher than the transcript level CV of lab-grown *A. thaliana* plants in a recent study (Cortijo et al., 2019)'. In the Cortijo study they use a corrected CV measurement - does that affect the comparison?

Additionally, the authors harvest plants on two separate days. They acknowledge that this is a source of additional heterogeneity (In the PCA in figure 1). It seems important to redo parts of the analysis on plants just from the same day - Although most plants were harvested on the first day, it was unclear to me how much of the variability was caused by the two separate timepoints being added together, which seems to somewhat go against the idea of the individual plant approach. An equivalent bulk study of plants a week older would be expected to have differences in gene expression compared to bulk measurements of younger plants.

The authors also find a spatial correlation between plants across the field and their transcriptomes and phenotypes. It wasn't clear to me what fraction of differences between Maize plants were due to extrinsic differences (Location, harvesting time, developmental stage?) versus 'intrinsic' differences in the levels of transcription or metabolites between plants. Is there a way of simply quantifying this, or making the current analysis clearer to the reader (as the authors have already done a great job of looking at each of these aspects individually)? This question of where the variability comes from seems to be one that would be of general interest to MSB readers.

References to authors previous work: At two points in the paper the authors refer to their previous work:

' gene expression variations among individual *Arabidopsis thaliana* plants grown under the same stringently controlled lab conditions contain a lot of information on the molecular wiring of the plants, on par with traditional expression profiles of pooled plant samples subject to controlled perturbations (Bhosale et al, 2013).' And

In previous work, we showed that expression variations among individual *Arabidopsis thaliana* plants, all grown under the same stringently controlled conditions, can efficiently predict gene functions (Bhosale et al., 2013)

From this description, readers will miss that the work in Bhosale et al. was analysis of individual *Arabidopsis* leaves from different genotypes grown in different labs. The first step of the work was to remove these differences 'in silico', but this is not mentioned here. Readers might think that these plants were all genetically identical plants grown under the exact same conditions in one laboratory, which is not the case. The description of this work should be revised to reflect the actual conditions used.

Reviewer #2:

In this manuscript, Cruz and colleagues put forward an intriguing concept to connect genotype to phenotype in plants, namely the correlation of gene expression and metabolome with phenotypes in field grown individuals. The reasoning is quite simple: functionally relevant and connected regulatory networks will respond coherently to the subtle variations in growth conditions encountered within a field. This in turn may provide the ideal level of variation within "omix"

datasets to filter out relevant genes for important real-world phenotypes. The authors present very thorough analyses of selected phenotypes along with transcriptome and metabolome data from roughly 60 plants picked from a small field site and convincingly demonstrate that the single plant approach works well and indeed represents a promising strategy for gene function discovery in the future.

Some of the major conclusions include that local variation within a field is pervasive, both at the phenotypic and the molecular level; that single point molecular data obtained from the field have at least as much information than pooled samples derived from controlled conditions, and that causal genes can be identified with a good probability for phenotypes that are connected with the tissue sample.

Overall, the manuscript is very well written, the analyses are carried out at a very high level and the conceptual advance is clear.

I only have the following points that I feel need to be addressed:

1. Figure 1 does not allow to appreciate the spatial correlations of plants within the sampling site. We need some sort of color coordinate system to recognize the patterns in the analyses.
2. Variable genes must be analyzed for circadian or diurnal effects, since the sampling has been carried out over a longer period of time. What variation is left when all genes with known diurnal expression are taken out of the analysis?
3. I think that the transcript-phenotype correlations deserve more attention. I would like to see more data on why certain phenotypes can be predicted and others not. The authors comment that genes expressed in the leaf may also be expressed elsewhere, e.g. in the embryo and therefore leaf transcriptome may predict embryo phenotypes. These types of correlations can systematically be tested from published data. Is the leaf transcriptome more similar to embryos than to flowers? Do they share specific important regulatory modules, e.g. auxin signaling, that may explain the connection? These types of analyses would provide important groundwork for future studies, since they would allow to design sampling strategies that maximize predictive power.
4. The authors advertise the metabolome, but essentially never make use of it for their analyses. I would like to learn more about this dataset and the correlation with the transcriptome. I would imagine that individual field grown plants could represent a gold mine to drill deep into this correlation, especially with regards to plant pathogen interactions.

Reviewer #3:

The study profiled phenotype and the transcriptome of 60 individual maize plants of the same inbred background (B104) in the same field to study gene-phenotype relationships, with 50 of them also with metabolite profiles. The idea is to use the inherent stochasticity of molecular processes and external factors (e.g. variability in the micro-environment) as perturbations to discover novel gene functions that could be ignored/averaged out in the pooled lab samples. The study showed that the average transcript level coefficient of variation (CV) in the field is three times higher than those for lab-grown *A. thaliana* plants. The variability allows the predictions of phenotypes from both transcriptome and metabolome data, which are comparable with each other and with lab data under controlled perturbation. However, the combination of both transcriptome and metabolome data does not outperform the model learned individually and the hypothesis is that the relevant

phenotype information is redundantly present in both data types. For gene function predictions, the single-plant dataset outperforms more than 75% of the sampled datasets for predict genes involved in leaf development. For novel gene functions, literature screening is used to complement the possible incompleteness in available GO annotations and some pieces of evidence are found to support the model. Both Elastic net and random forest techniques were used to construct models predicting the phenotypes of individual plants as a function of the transcript and metabolite levels. The function of maize genes are predicted from the function of their coexpression network neighbors ('guilt-by-association') via a command-line version of PiNGO.

This is an interesting study for revealing gene functions from subtle phenotypical variations. In some sense, this might complement the usual harsh perturbation based experiment design. However, the novel gene functions are not confirmed experimentally. Literature screening provides some evidence for a small percentage of target genes but not all. On the other hand, could the variation be due to somatic mutations? Since there is RNA-Seq data available for individual plants, it could be checked to provide further clues for the target genes.

Reviewer #1:

In the work of Cruz et al., The authors take an interesting approach, examining the transcriptomes and metabolomes of individually grown maize plants in the field, and comparing these results to the individual phenotypes of the plants. In general, this appears a promising approach to understanding and transferring lab based plant results to the field. It reveals a high level of phenotypic variability between plants in the field which can to some degree be matched to transcriptional differences between plants. In general, I think the paper is an interesting and important step forward for the plant systems biology field. I have the following concerns:

Comment 1.1: Readability for a general audience: Many of the aspects of this study will be very interesting to the general audience of Molecular Systems Biology. However, it is written for a plant audience currently. It would aid the impact of the work if the introduction and discussion could be a little less plant specific. The issue of individual variability within populations has been explored in microbial and mammalian systems, and I would think that the MSB readers would appreciate a link to this work, as well as more general points to be made about the use of using the variability between individuals under the same environment for generating regulatory networks.

Reply 1.1: We extended the introduction to include a discussion of previous work on individual variability in other kingdoms of life, noting that this work is mostly focused on either studying developmental plasticity of organisms (Waddington's canalization concept) or studying stochasticity in single-celled organisms or single cells of multicellular organisms. At the end of the discussion, we now highlight that systems biology studies in most organisms moved straight from replicated experiments on pools of individuals to single-cell profiling, and that the 'individual' level deserves a reappraisal.

Comment 1.2: Analysis of Individual transcriptome data. In examining individual transcriptional variability (either single cell or single individual) it is important to correct for technical noise (e.g, <https://doi.org/10.1038/nmeth.2645>). As detected levels get lower, CV increases due to this. Hence, it is important to use, or consider, a corrected CV score to account for this. I can't see a figure showing how CV scales with the mean expression of genes in the paper. It would be interesting if this could be added, as well as a more detailed explanation for why CV was chosen as the measure of variability.

Additionally, the authors write 'The average transcript level CV of ~0.3 is about three times higher than the transcript level CV of lab-grown *A. thaliana* plants in a recent study (Cortijo et al., 2019)'. In the Cortijo study they use a corrected CV measurement - does that affect the comparison?

Reply 1.2: We did indeed not take into account previously that the CV increases for lowly expressed genes due to technical noise, and now calculate normalized CV values instead as in Cortijo et al. (2019). This also affected our ranking of variable genes, with the main effect that some lowly expressed genes, including several histones that previously showed behavior different from other chromatin-associated genes, disappeared from the list of most variable genes. The sentence in the previous version of our manuscript on the transcript level CV in our data being about 3 times higher than in the Cortijo et al. (2019) study was based on visual comparison of the $\log(\text{CV}^2)$ values in our data with figure 1B in Cortijo et al. (2019), which displays the log-scale uncorrected CV^2 distribution as a function of mean expression. We now computationally compared our dataset (without day-of-harvest, sequencing batch and population structure effects, so with somewhat lower variance) to the Cortijo et al. (2019) dataset, and found that the average transcript level CV in our dataset is 2.49 times higher than in the Cortijo et al. (2019) dataset. Plots comparing the CV^2 values and their relationship to mean expression in both our dataset and the Cortijo et al. (2019) dataset can be found in the new **Appendix Figure S5**.

Comment 1.3: Additionally, the authors harvest plants on two separate days. They acknowledge that this is a source of additional heterogeneity (In the PCA in figure 1). It seems important to redo parts of the analysis on plants just from the same day - Although most plants were harvested on the first day, it was unclear to me how much of the variability was caused by the two separate timepoints being added together, which seems to somewhat go against the idea of the individual plant approach. An equivalent bulk study of plants a week older would be expected to have differences in gene expression compared to bulk measurements of younger plants.

Reply 1.3: This is a very good point. We initially left the day-of-harvest (DOH) effect in the data because it gives rise to variation that may be useful from the perspective of predicting gene functions or phenotypes, but this indeed undermines the single-plant character of our study. We therefore decided to remove the DOH effect (together with a sequencing batch effect we did not adequately correct for previously and a genetic population structure effect that we discovered while addressing a comment of Reviewer #3, see the introductory paragraph and our response to **Comment 3.2**), and redid all analyses on the corrected dataset. The DOH effect (in contrast to the batch and SNP effects) was found to affect a substantial proportion of variables, notably 22.2% of transcripts, in particular of genes involved in photosynthesis, nucleosome organization and transcriptional regulation (see **Appendix Table S1** and **Dataset EV3**). Removing the DOH effect removed some 8% of the variability for the average transcript, about 5% of the variability for the average metabolite, and between 0 and 20% of the phenotype variability, most notably for leaf 16 blade length (18.5% decrease in variation, see **Figure EV1**). However, our downstream analyses still generate comparable results overall, and our conclusions remain the same.

Comment 1.4: The authors also find a spatial correlation between plants across the field and their transcriptomes and phenotypes. It wasn't clear to me what fraction of

differences between Maize plants were due to extrinsic differences (Location, harvesting time, developmental stage?) versus 'intrinsic' differences in the levels of transcription or metabolites between plants. Is there a way of simply quantifying this, or making the current analysis clearer to the reader (as the authors have already done a great job of looking at each of these aspects individually)? This question of where the variability comes from seems to be one that would be of general interest to MSB readers.

Reply 1.4: As part of our data reanalysis, we used linear mixed effect (LME) models with harvesting date (DOH), population substructure (based on SNPs) and sequencing batch as fixed effects and incorporating the spatial structure of the field setup in the error model. From this, we estimated how much of the variance in all variables was due to these effects (see **Figure EV1**). Plants were harvested based on identical developmental stage and minor developmental differences were not scored, so we cannot account for these. Overall, the batch, DOH and SNP effects explained only minor proportions of the variance for most genes and metabolites (see **Figure EV1**). Additionally, 14.1% of the transcript profiles and 8% of the metabolite profiles were found to have a spatial structure as judged by Moran's I ($q \leq 0.01$), and around 60% on average of the variance in those transcript and metabolite profiles (after removing DOH, SNP and batch effects) was due to spatial covariance according to the LME modeling results (see **Appendix Figure S6**). There is no guarantee however that the remaining ~40% of the variability for the average transcript or metabolite (the i.i.d. distributed part of the LME model residuals) is intrinsic. A substantial portion of what is left may also be extrinsic in the sense that it may be due to environmental differences without spatial structure or with a spatial structure on a smaller scale than can be observed in our sampling grid.

Comment 1.5: References to authors previous work: At two points in the paper the authors refer to their previous work: 'gene expression variations among individual *Arabidopsis thaliana* plants grown under the same stringently controlled lab conditions contain a lot of information on the molecular wiring of the plants, on par with traditional expression profiles of pooled plant samples subject to controlled perturbations (Bhosale et al, 2013).' And

In previous work, we showed that expression variations among individual *Arabidopsis thaliana* plants, all grown under the same stringently controlled conditions, can efficiently predict gene functions (Bhosale et al., 2013)

From this description, readers will miss that the work in Bhosale et al. was analysis of individual *Arabidopsis* leaves from different genotypes grown in different labs. The first step of the work was to remove these differences 'in silico', but this is not mentioned here. Readers might think that these plants were all genetically identical plants grown under the exact same conditions in one laboratory, which is not the case. The description of this work should be revised to reflect the actual conditions used.

Reply 1.5: The reviewer is correct that insufficient details were given on the experimental conditions used to generate the *Arabidopsis* data we analyzed previously in Bhosale et al. (2013), and that this could cause misunderstandings. This has been corrected. We opted to not include these details in the introduction, to avoid breaking the flow of the text, but include them instead in the results section upon second

mention of the Bhosale et al. (2013) study. This allows us to provide more context and better highlight similarities and differences with the present study.

Reviewer #2:

In this manuscript, Cruz and colleagues put forward an intriguing concept to connect genotype to phenotype in plants, namely the correlation of gene expression and metabolome with phenotypes in field grown individuals. The reasoning is quite simple: functionally relevant and connected regulatory networks will respond coherently to the subtle variations in growth conditions encountered within a field. This in turn may provide the ideal level of variation within "omix" datasets to filter out relevant genes for important real-world phenotypes. The authors present very thorough analyses of selected phenotypes along with transcriptome and metabolome data from roughly 60 plants picked from a small field site and convincingly demonstrate that the single plant approach works well and indeed represents a promising strategy for gene function discovery in the future. Some of the major conclusions include that local variation within a field is pervasive, both at the phenotypic and the molecular level; that single point molecular data obtained from the field have at least as much information than pooled samples derived from controlled conditions, and that causal genes can be identified with a good probability for phenotypes that are connected with the tissue sample. Overall, the manuscript is very well written, the analyses are carried out at a very high level and the conceptual advance is clear.

I only have the following points that I feel need to be addressed:

Comment 2.1: Figure 1 does not allow to appreciate the spatial correlations of plants within the sampling site. We need some sort of color coordinate system to recognize the patterns in the analyses.

Reply 2.1: As incorporating additional colors in the new Figure 1 led to a messy figure (because also the harvesting day, genetic population structure and sequencing batch groups are indicated there on the PCA), we now present PCA plots with color-coded field locations in **Appendix Figure S3**.

Comment 2.2: Variable genes must be analyzed for circadian or diurnal effects, since the sampling has been carried out over a longer period of time. What variation is left when all genes with known diurnal expression are taken out of the analysis?

Reply 2.2: We tried to avoid diurnal effects as much as possible by keeping the sampling timeframe as short as possible. All plants and organs were sampled in a 2-hour timeframe, and the same timeframe (10:00 am to 12:00 pm) was used on both harvest days, so we expected diurnal effects to be limited. Even in lab experiments, sampling often takes longer than 2 hours and diurnal corrections are to our knowledge rarely if ever performed.

Even so, it is indeed possible that part of the expression variation between plants is caused by diurnal effects in the 2-hour sampling period. Removing genes with known diurnal variation patterns will not teach us a lot about what fraction of a transcript's variance is caused by diurnal effects, as this will not impact the variation of the

remaining genes. On the other hand, fully estimating and correcting for diurnal variation effects in our data is not evident as the exact time points at which the plants were harvested were not recorded. In theory, it should be possible to use a latent variable model to estimate the unknown diurnal phases of samples using the expression of known diurnal rhythm marker genes, and then remove the estimated effects. This is for instance also done for removing cell cycle effects in single-cell RNA-seq datasets (see e.g. Buettner et al. 2015 Nature Biotechnology, <https://doi.org/10.1038/nbt.3102>). The efficacy of such an approach would however likely be limited given that the sampling period in our data is only 2 hours long.

We took an alternative approach to assess which proportion of transcript variance could be due to diurnal effects. We compared the diurnal variations for maize transcripts as observed in a recent study (Lai et al. BMC Genomics (2020) 21:428, <https://doi.org/10.1186/s12864-020-06824-3>) with the corresponding transcript level variations among the individual plants in our study. We first preprocessed and normalized the Lai et al. (2020) raw data using the same protocol as used for our data, except that read mapping was not done with GSNAP, but with the faster HiSat2. The 5% most lowly expressed genes in both datasets were removed, and we then compared plots of the transcript CV² versus mean expression for the two datasets (**Figure R1** below). 9,329 out of 18,171 transcripts in the single-plant dataset were identified by Lai et al. (2020) as diurnally varying at $q < 0.05$ (Table S3 in Lai et al. 2020), and after removing the 5% most lowly expressed genes, 2,256 transcripts showed a strong diurnal rhythm ($q < 1e-05$). We mainly focused our analysis on these transcripts (colored dots in **Figure R1**). A couple of observations can be made from **Figure R1**. First, genes with higher mean expression in the Lai et al. (2020) dataset tend to also be more highly expressed on average in the single-plant dataset. This renders the CV² versus mean expression plots for the two datasets more comparable. For highly expressed genes, the CV² trend values in the single-plant dataset are only about 19.1% of the CV² trend values in the Lai et al. (2020) dataset, indicating that the expression variability in the single-plant data is generally substantially lower than the variability of genes over a 24-hour period. CV² values in both datasets only become similar for lowly expressed genes, for which CV values are dominated by technical noise.

That there might be some time-of-day effect in our data is indicated by the fact that strongly rhythmic genes in the Lai et al. (2020) dataset have a higher median normalized CV (i.e. $\log_2(\text{CV}^2/\text{trend})$, see Methods) in our data than weakly rhythmic or non-rhythmic genes (Mann-Whitney U test p values $< 1e-67$, **Figure R2**). The set of strongly rhythmic genes is enriched in genes involved in photosynthesis and small-molecule metabolism (**Dataset EV3**). The shift in median normalized CV between strongly rhythmic genes and other genes is however small compared to the range of normalized CV values across all genes (**Figure R2**), indicating that only a minor part of the expression variance in our dataset is due to diurnal effects. Furthermore, it cannot be excluded that there are other reasons or cues than diurnal rhythmicity that may cause strongly rhythmic genes to be more variably expressed in our dataset than the average gene. The difference in median normalized CV between weakly rhythmic and non-rhythmic genes is not significant (Mann-Whitney U test $p = 0.7523$).

We also assessed whether diurnally varying genes would be differentially affected in our data dependent on their time of peak expression in the Lai et al. (2020) dataset.

Figure R3C and **Table R1** show that rhythmic genes with peak expression between 9:00 and 13:00 tend to have higher normalized CV values in the single-plant dataset, which could be taken to suggest that these genes show considerable variation in the 10:00-12:00 harvesting timeframe. On the other hand, these genes also have higher normalized CV values in the Lai et al. (2020) dataset (**Figure R3B**), indicating that they are more variably expressed overall than rhythmic genes peaking at other times of the day. Generally, there are no obvious shifts in the normalized CV of genes peaking at any particular time of day in the single plant dataset versus the Lai et al. (2020) dataset (**Figure R3**), indicating that any time-of-day effects in our dataset similarly affect genes peaking at different times of day.

In summary, up to a few thousand genes may exhibit minor diurnal variation effects because of the 2-hour sampling timespan of our experiment. These variations however do not disturb the single-plant character of the study (in contrast to e.g. the day-of-harvest effect), and we therefore did not attempt to remove them. Moreover, such removal would likely work only partially and would require estimating diurnal effect sizes simultaneously with the DOH, sequencing batch, population substructure and particularly spatial autocorrelation effects, which is technically challenging. In order not to inflate the manuscript size too much, we only included the analysis depicted on **Figure R2** in the new manuscript version (**Appendix Figure S4**).

Table R1. Median normalized CV² values in the single-plant dataset of rhythmic genes according to peak expression time. Median normalized CV² values are calculated from the single-plant expression data for groups of rhythmic genes ($q < 5e-02$) with peak expression in a given 2-hour interval.

peak expr. time	09:00- 11:00	11:00- 13:00	13:00- 15:00	15:00- 17:00	17:00- 19:00	19:00- 21:00	21:00- 23:00	23:00- 01:00	01:00- 03:00	03:00- 05:00	05:00- 07:00	07:00- 09:00
median CV ²	-0.264	-0.229	-1.157	-1.217	-1.459	-1.721	-1.921	-1.904	-1.791	-1.398	-0.751	-0.491

Figure R1. Comparison of the CV^2 distributions of transcripts in the single-plant data and the Lai et al. (2020) dataset profiling diurnal rhythms. (A) Plot comparing the CV^2 values in both dataset for genes identified in Lai et al. (2020) as having a strong diurnal rhythm ($q < 1e-5$). The pink line displays a linear fit of the datapoints in log space, points on the black line have equal CV^2 in both datasets. **(B)** CV^2 versus mean expression plot for the Lai et al. (2020) dataset. The red trendline is a generalized linear model (GLM) fit of the gamma family with identity link, as in Brennecke et al. (2013) **(C)** CV^2 versus mean expression plot for the single-plant dataset, with a trendline in orange constructed as for panel (B). **(D)** Overlay of the CV^2 versus mean expression relationships in both datasets, for genes with a strong diurnal rhythm ($q < 1e-5$) in Lai et al. (2020). In all plots, genes with a strong diurnal rhythm ($q < 1e-5$) are colored according to their mean expression in the Lai et al. (2020) dataset (see color legend on the right).

Figure R2. Normalized expression CV distributions in the single-plant dataset for diurnally varying genes versus non-diurnally varying genes. Violin plots of normalized CV distributions are shown for genes identified in Lai et al. (2020) as strongly rhythmic (A), weakly rhythmic (B) or non-rhythmic (C), and for all genes (D).

Figure R3. Time of peak expression mapped to CV^2 distributions of the single-plant data and the Lai et al. (2020) dataset. (A) Plot comparing the CV^2 values in both dataset for genes identified in Lai et al. (2020) as having a strong diurnal rhythm ($q < 1e-5$). The pink line displays a linear fit of the datapoints in log space, points on the black line have equal CV^2 in both datasets. (B) CV^2 versus mean expression plot for the Lai et al. (2020) dataset. The red trendline is a generalized linear model (GLM) fit of the gamma family with identity link, as in Brennecke et al. (2013) (C) CV^2 versus mean expression plot for the single-plant dataset, with a trendline in orange constructed as for panel (B). (D) Overlay of the CV^2 versus mean expression relationships in both datasets, for genes with a strong diurnal rhythm ($q < 1e-5$) in Lai et al. (2020). In all plots, genes with a strong diurnal rhythm ($q < 1e-5$) are colored according to their time of peak expression in the Lai et al. (2020) dataset (see color legend on the right).

Comment 2.3: I think that the transcript-phenotype correlations deserve more attention. I would like to see more data on why certain phenotypes can be predicted and others not. The authors comment that genes expressed in the leaf may also be expressed elsewhere, e.g. in the embryo and therefore leaf transcriptome may predict embryo phenotypes. These types of correlations can systematically be tested from published data. Is the leaf transcriptome more similar to embryos than to flowers? Do

they share specific important regulatory modules, e.g. auxin signaling, that may explain the connection? These types of analyses would provide important groundwork for future studies, since they would allow to design sampling strategies that maximize predictive power.

Reply 2.3: We assume this question of Reviewer #2 is about our gene function prediction performance results, where performance scores of our dataset relative to the datasets sampled from the SRA database are better for some organ development categories than for others. In the previous version of the manuscript, we found that our dataset outperformed the datasets sampled from SRA for predicting genes involved in embryo, root and leaf development, despite the fact that all datasets only profiled leaves. It is important to stress however that this does not mean that our dataset predicts e.g. root development genes well, only that it does so better than the SRA datasets. For actual data on how well genes involved in a particular process are predicted, one needs to look at the recall, precision and F-measure score plots, now given in **Dataset EV12**. The old prediction performance figure for root development for instance, reproduced below as **Figure R4**, shows that recall, precision and F-measure are in fact low for all datasets, or in other words that root development genes are not at all well predicted. Based on our corrected dataset (after removal of DOH, population structure and sequencing batch effects), the same is true for flower development, and, surprisingly, also leaf development, despite the fact that it scores very well relative to the SRA datasets (**Dataset EV12**, root development could no longer be scored because it produced too few root development predictions, but this can in fact be considered even worse). The F-measure values for embryo, seed and fruit development on the other hand go up to ~ 0.08 , which is substantially higher than for e.g. leaf development (~ 0.01). The cause for this is technical rather than biological: fewer genes are annotated in GO to leaf, root or flower development than to embryo, seed and fruit development (**Table R2** below), likely because the latter processes have been studied and annotated more than the former (also in *Arabidopsis thaliana*, where most of the GO annotations for plants derive from). This means there are more known genes for e.g. embryo development than for leaf development in the network. As the number of known genes for a process increases, also the density of genes annotated to this process will increase in certain parts of the network, assuming that genes functioning in the same process will cluster together more than expected by chance (guilt-by-association dogma). This in turn makes it statistically easier to associate genes (either known or new) with the process concerned, which explains why prediction performance F-measures for embryo development can be higher than for leaf development despite the fact that leaves and not embryos were profiled.

For this reason, our function prediction performance results can better be compared across datasets (e.g. our network versus the sampled SRA networks) than directly across GO categories. These relative performances can then still be interpreted across GO categories however. For instance, based on our corrected dataset, the single-plant network scores very well relative to the SRA networks for leaf and embryo development (**Figure 6**), but worse for other developmental processes. We consider this a positive result as leaves and embryos (which contain embryonic leaves) are the two developmental processes that are most closely related to the material profiled, both in the single plant and the SRA datasets.

Table R2. Numbers of genes annotated to different developmental process categories in the maize GO annotation.

GO ID	GO Name	Number of annotated genes
48366	leaf development	508
9790	embryo development	707
48316	seed development	907
9908	flower development	639
48364	root development	653
10154	fruit development	959

The question remains why it is possible to predict e.g. seed development genes from leaf data. As already stated in the previous version of the manuscript, it is known that several genes expressed in leaves have roles in the development of other organs. Parts of e.g. the auxin signaling network and essentially all other hormone signaling pathways are reused in different developmental programs. A thorough analysis of which parts of which pathways are conserved across which organs is outside the scope of this study. It is however very likely that many commonalities between different developmental programs remain to be discovered and/or annotated.

Figure R4. Gene function prediction performance plots for root development in the previous version of the manuscript.

Comment 2.4: The authors advertise the metabolome, but essentially never make use of it for their analyses. I would like to learn more about this dataset and the correlation with the transcriptome. I would imagine that individual field grown plants could represent a gold mine to drill deep into this correlation, especially with regards to plant pathogen interactions.

Reply 2.4: We did indeed focus mostly on interpretation of the results of transcriptome analyses, mainly because of the fact that many metabolites in our dataset remain unknown and hence difficult to interpret. Although we did use the metabolome data for various analyses in the previous manuscript version, e.g. PCA analysis, spatial autocorrelation analysis and predictive models for the phenotypes, we acknowledge that we could have done more. In the revised manuscript, we also included clustering results for autocorrelated metabolite profiles and associations of these clusters with phenotypes (**Datasets EV4, EV5 and EV6**), a ranking of metabolites according to field variability (**Appendix Figure S7 and Dataset EV7**) and spatially corrected correlation analyses to associate single metabolites to phenotypes (**Dataset EV15**), but the results remain difficult to interpret biologically. Correlations between transcript and metabolite profiles are depicted in the global hierarchical clustering in **Figure EV4**. Which metabolites and transcripts belong to which cluster can be found in **Dataset EV9**. We can unfortunately not go into detail about which metabolites are linked to genes involved in various kinds of plant-pathogen interactions, as this is not our field of expertise. All our datasets are made available however for further mining, and we definitely agree that many other interesting findings may emerge from the data if they are looked at from another perspective than ours.

Reviewer #3:

The study profiled phenotype and the transcriptome of 60 individual maize plants of the same inbred background (B104) in the same field to study gene-phenotype relationships, with 50 of them also with metabolite profiles. The idea is to use the inherent stochasticity of molecular processes and external factors (e.g. variability in the micro-environment) as perturbations to discover novel gene functions that could be ignored/averaged out in the pooled lab samples. The study showed that the average transcript level coefficient of variation (CV) in the field is three times higher than those for lab-grown *A. thaliana* plants. The variability allows the predictions of phenotypes from both transcriptome and metabolome data, which are comparable with each other and with lab data under controlled perturbation. However, the combination of both transcriptome and metabolome data does not outperform the model learned individually and the hypothesis is that the relevant phenotype information is redundantly present in both data types. For gene function predictions, the single-plant dataset outperforms more than 75% of the sampled datasets for predict genes involved in leaf development. For novel gene functions, literature screening is used to complement the possible incompleteness in available GO annotations and some pieces of evidence are found to support the model. Both Elastic net and random forest techniques were used to construct models predicting the phenotypes of individual plants as a function of the transcript and metabolite levels. The function of maize genes are predicted from the function of their coexpression

network neighbors ('guilt-by-association') via a command-line version of PiNGO.

Comment 3.1: This is an interesting study for revealing gene functions from subtle phenotypical variations. In some sense, this might complement the usual harsh perturbation based experiment design. However, the novel gene functions are not confirmed experimentally. Literature screening provides some evidence for a small percentage of target genes but not all.

Reply 3.1: Reviewer #3 is correct that experimental validations of our gene function predictions are lacking. We do plan follow-up experiments in particular for the top genes predicted to influence leaf length and width, but it would still cost us about a year to generate the necessary lines, and we would like to publish the results we already have before that time. Although the literature screening we performed only gives partial information on the validity of our predictions, it does give a good idea about whether or not our predictions make sense. Using a corrected dataset (with day-of-harvest, population structure and sequencing batch effects removed, see the introductory paragraph), we redid the gene function predictions and literature validations and found that the percentage of predictions validated to at least some extent by literature increased to more than 45%.

Comment 3.2: On the other hand, could the variation be due to somatic mutations? Since there is RNA-Seq data available for individual plants, it could be checked to provide further clues for the target genes.

Reply 3.2: There is indeed residual genetic variability in the B104 inbred line due to germline and somatic mutations, and how much of the molecular and phenotypic variability between plants can be explained by this genetic variability is a very interesting question. We therefore investigated the genetic variant structure of our RNA-seq data, and unexpectedly discovered that two slightly different subpopulations of plants were part of our experiment, likely derived from different mother plants (**Appendix Figure S1**). The SNPs differentiating the subpopulations are found on all chromosomes, but mainly in a few regions on chromosome 1 (**Appendix Figure S2**).

Because this introduces a systematic effect that may interfere with the single-plant character of our study, we decided to remove this population structure effect using linear mixed effect (LME) modeling, together with the day-of-harvest and sequencing batch effects (see introduction). The levels of 103 transcripts, 1 metabolite and none of the phenotypes were found to exhibit a significant population structure effect ($q \leq 0.01$ in the LME models, **Appendix Table S1**), which indicates that the effects of the population structure on the molecular and phenotypic data are limited, even before correcting for them.

After removal of the population structure effect, there are however still a lot of other genetic variants in the data that may affect transcript, metabolite or phenotype levels. In total 10,311 biallelic SNPs (including 1,377 SNPs for which the allele profile corresponded with the population structure) were called in at least 48 plants and had a minor allele frequency (MAF) of $\geq 5\%$ (≥ 3 individuals) after imputation of missing values (**Appendix Table S6**). The number of other types of variants was comparatively limited. We focused on this set of 10,311 biallelic SNPs to investigate whether genetic variants significantly affect the phenotypes of the individual plants.

This analysis is necessarily incomplete, as we cannot recover e.g. SNPs in cis-regulatory regions from the RNA-seq data, but it should give a good idea about the distribution of effect sizes of genetic variants in our dataset. The effects of many cis-regulatory variants are likely also captured indirectly through linkage with variants in the coding sequence. We focused on linking genetic variability to phenotypic variability, and took two different approaches to estimate the effect of biallelic SNPs on the phenotypes observed.

In a first approach, we performed a classical GWAS analysis to associate SNPs with phenotypes. We focused on a set of 10,311 biallelic SNPs that were called in at least 48 plants and that had a minor allele frequency (MAF) after imputation of missing values of $\geq 5\%$ (≥ 3 individuals). We did not uncover any reliable evidence linking SNPs to phenotypic differences. A single SNP, which is not associated with a known gene, was found to surpass the significance threshold (Bonferroni-corrected $p \leq 0.01$) for ear length, but the corresponding quantile-quantile (Q-Q) plot displays abnormalities indicating that this result is likely unreliable (**Appendix Figure S9**).

In parallel, we used the biallelic SNPs as features in machine learning models to predict the phenotypes, similar to the analyses we did for linking transcriptome and metabolome data to phenotypes. This gives us a view on how much information the set of SNPs as a whole contains about any particular phenotype (rather than how much information individual SNPs contain). An added benefit of the machine learning approach is that the models are trained using crossvalidation and tested on out-of-bag samples, giving a more accurate view on their predictive power on unseen samples than in a classical GWAS. Also in the machine learning models, SNPs were not found to be good predictors of the individual plant phenotypes (**Appendix Table S7**). Together, these results show that the impact of the residual genetic variance in our population on phenotypes is small.

Given these results, it made little sense to us to also perform a full-scale e-GWAS analysis to check the effects of genetic variants on gene expression levels. We did however investigate how genetic variability compares with micro-environmental or stochastic variability as an information source for gene function prediction. To this end, we compared how the function prediction performance of the expression variability in our dataset (individuals of B104 inbred background) compares to that of the expression variability in a genetic diversity panel, using the same sampling methodology we used on data from the Sequence Read Archive (SRA) on a dataset from Kremling et al. 2018 (<https://doi.org/10.1038/nature25966>). We found that the gene function prediction performance (F -measures) of the single-plant dataset was higher than that of all sampled diversity datasets (**Figure EV5**), indicating that individual plant variation is at least as valuable as genetic variation for unraveling gene function.

Thank you for sending us your revised manuscript. We have now heard back from two of the three reviewers who were asked to evaluate your study. Unfortunately, after a series of reminders we did not manage to obtain a report from Reviewer #2. In the interest of time, and since the other two reviewers' recommendations are quite similar, I prefer to make a decision now rather than further delaying the process. As you will see the reviewers are satisfied with the modifications made and think that the study is now suitable for publication.

Before we can formally accept your manuscript, we would ask you to address a few remaining editorial issues listed below.

REFEREE REPORTS

Reviewer #1:

The authors have really taken on board the comments of the reviewers and have carried out an impressively thorough update of the manuscript. I am happy to recommend publication.

Reviewer #3:

The manuscript has been substantially improved in this revision. Its great that the population structure effect could be stratified with the variant data, which further improved the analysis results.

The Authors have made the requested editorial changes.

Accepted**17th Nov 2020**

Thank you again for sending us your revised manuscript. We are now satisfied with the modifications made and I am pleased to inform you that your paper has been accepted for publication.

Corresponding Author Name: Steven Maere

Manuscript Number: MSB-20-9667